# Towards Understanding Metacognition in Large Reasoning Models

## Abstract

Large Reasoning Models (LRMs) have demonstrated remarkable capabilities on complex tasks. Despite these advances, we identify a fundamental limitation: current LRMs impose fixed cognition patterns, lacking the intrinsic ability to be aware of, or regulate their own reasoning processes. This signifies a critical absence of metacognition—an essential faculty in human intelligence. Building on psychology and cognitive science, we first construct a functional framework for metacognition in LRMs, separating internal informational signals from behavioral abilities. This framework is then applied to a comprehensive investigation on seven state-of-the-art LRMs and reveals a consistent gap: while metacognitive information is present and predictive, it often fails to translate into reliable monitoring or control behaviors. To address this gap, we introduce two distinct paradigms for instilling metacognition in LRMs: (1) an emergent approach that leverages prompting to orchestrate metacognitive functions, such as task assessment, confidence monitoring, and strategy regulation; (2) an intrinsic approach that internalizes these faculties by encoding structured meta-cognitive information directly into the model's parameters through training. Overall, our results indicate that integrating metacognitive reasoning improves task performance and offers a valuable lens for the design of future reasoning models.

## 1 Introduction

Large Reasoning Models (LRMs) like OpenAI-o1 (Jaech et al., 2024) and Deepseek-R1 (Guo et al., 2025a) have achieved remarkable success in complex domains such as coding and mathematics. At first glance, these models appear to exhibit advanced, reflective behaviors within their long chain-of-thought (CoT) (Wei et al., 2022) reasoning. However, a closer look reveals a potential fragility in their cognitive processes. For instance, when tasked with solving a complex-variable equation under the explicit constraint that "$z$ is a positive real number," an LRM may persist in a fixed reasoning pattern like "the problem likely intended for $z$ to be a complex number," a phenomenon termed *reasoning rigidity* (Jang et al., 2025; Araya, 2025). Furthermore, current LRMs also frequently display illusory self-correction, employing introspective phrases like "Wait, let me double-check..." without any actual adjustment to a flawed reasoning trajectory (Guo et al., 2025a; Wang et al., 2025b). These consistent failures in self-monitoring and adaptation reveal a core limitation of current LRMs.

We argue that this deficit can be productively framed as an absence of metacognition (Ackerman & Thompson, 2017; Norman et al., 2019; Tankelevitch et al., 2024)—the ability to monitor and control one's own cognitive processes. In the theory of human cognition, metacognition is essential for evaluating potential reasoning errors (Yeung & Summerfield, 2012), calibrating uncertainty in decision-making (Qiu et al., 2018), and dynamically adapting strategies based on performance (Cary & Reder, 2002). This comparison to human intelligence raises a pivotal question for AI: do current LRMs possess any analogous capabilities, and if so, are they functionally engaged during inference? In this paper, we present the first systematic investigation into this fundamental question.

To structure this investigation, we introduce a functional framework for LRM metacognition, inspired by foundational models in cognitive science (Efklides & Misailidi, 2010; Ackerman & Thompson, 2017). Our framework decomposes metacognition into two components: **information** and **abilities**. Metacognitive information (Dayan, 2023; Norman et al., 2019), the basis for judgment, which includes both static knowledge (e.g., learned strategies in parameters) and dynamic ex-

perience (e.g., internal computational signals). Metacognitive abilities (Nelson & Dunlosky, 1991; Fiedler et al., 2019), the actions taken upon this information, which include *monitoring* (e.g., assessing task difficulty and confidence) and *control* (e.g., selecting reasoning strategies or decomposing problems). By deconstructing metacognition into these components, our framework provides a principled foundation to systematically probe whether, and in what way, metacognition emerges in contemporary LRMs.

Our investigation begins by empirically grounding the first component of our framework: **metacognitive information**. Focusing on the dynamic aspect of *experience*, we probe open-source LRMs to determine whether internal computational signals correlate with reasoning outcomes (§ 3). Our analysis yields a striking finding: signals spanning the entire Transformer architecture—from input-layer attributions to final-layer token probabilities—are highly predictive of answer success. Critically, we demonstrate that correct and incorrect reasoning traces generate statistically distinguishable internal signatures, providing the first empirical evidence that a machine-readable basis for metacognitive experience exists within these models.

This informational foundation compels the subsequent question: do state-of-the-art LRMs functionally leverage this information as observable metacognitive abilities (§4). Our evaluation of current leading reasoning models across a series of monitoring and control tasks reveals consistent failures. Specifically, we find that the models systematically misjudge task difficulty, display poorly calibrated confidence, and lack proactive planning and strategic flexibility. This evidence suggests that, while predictive metacognitive information may exist internally, it does not reliably translate into effective monitoring or control, exposing a critical gap in the capabilities of current LRM.

To bridge this information-to-ability gap, we propose two complementary paradigms for enhancing LRM metacognition: ❶ **Emergent Metacognition**, scaffolds this pathway at inference time through prompt-guided role-playing. We assign distinct metacognitive roles—such as 'Planner', 'Solver', and 'Verifier'—to simulate a complete monitoring and control loop. This external scaffolding forces the model to act on its latent experiences, effectively eliciting robust metacognitive behaviors without any parameter updates. ❷ **Internalized Metacognition**, directly enriches model's metacognitive *knowledge* through fine-tuning. We construct a dataset with explicit metacognitive annotations (e.g., plans, self-corrections) and fine-tune the model using a hybrid learning objective, directly embedding these capabilities into its parameters. Together, these two paradigms provide a comprehensive roadmap toward more introspective and reliable reasoning systems.

In summary, our findings reveal a clear dissociation between internal metacognitive information and externally observable metacognitive ability in current LRMs. This gap illuminates a new frontier for research: designing systems that not only possess self-awareness but can also act upon it, effectively bridging latent experience with adaptive behavior.

## 2 DEFINING METACOGNITION IN LRMS

Metacognition, first conceptualized in developmental psychology, refers to the capacity to monitor and control one's own cognitive processes (Flavell, 1979). According to the well-established two-level model of Nelson and Narens (Nelson & Dunlosky, 1991), metacognition comprises a *object-level* cognition (the act of thinking, perceiving, or remembering) and *meta-level* cognition (the act of thinking about one's thinking). While Large Reasoning Models (LRMs) do not possess subjective consciousness, their complex, multi-step reasoning processes create the functional necessity for such meta-level oversight. We therefore adopt a **functionalist perspective**: we investigate whether LRMs can exhibit behaviors and leverage internal signals that are functionally equivalent to human metacognition, enabling them to produce more reliable and robust reasoning.

Following established frameworks in cognitive science (Tankelevitch et al., 2024), we structure our functional model of LRM metacognition into two core components: Information and Abilities.

**Metacognitive Information** serves as the basis for judgment. It comprises (i) static *knowledge*—the latent understanding of tasks, strategies, and its own capabilities implicitly encoded in its parameters—and (ii) dynamic *experience*, which we operationalize as the internal computational signals (e.g., token probabilities) generated during a reasoning trace, serving as a functional analogue to a human's 'feeling of error'.

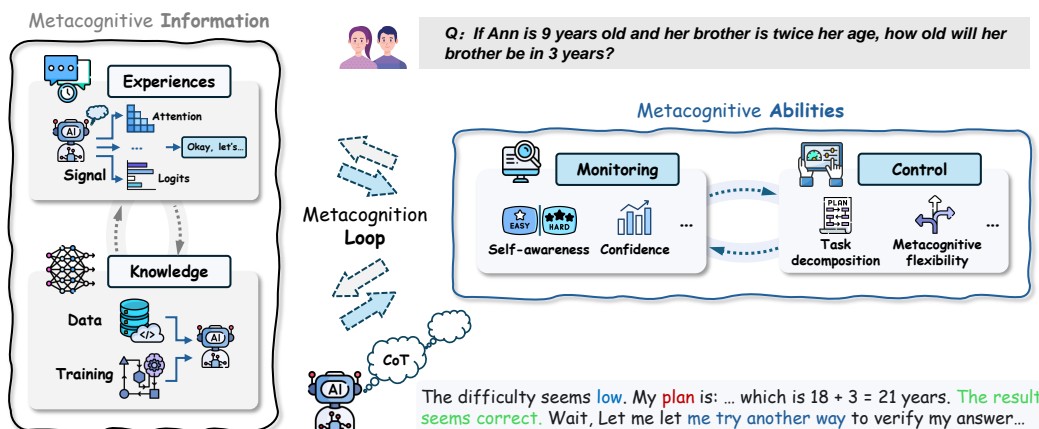

Figure 1: A functional framework for LRM metacognition.

**Metacognitive Abilities** are the actions taken based on this information. They consist of (i) *monitoring*, the capacity to generate self-assessments about its cognitive state or the task at hand, such as evaluating problem difficulty (§4.1) or estimating its confidence (§4.2); and (ii) *control*, the capacity to strategically alter its reasoning process, such as by performing task decomposition (§4.3) or exhibiting cognitive flexibility when encountering errors (§4.4).

This framework provides a structured lens through which we can systematically investigate the nascent metacognitive capabilities of modern LRMs (i.e., Fig. 1).

## 3 METACOGNITIVE INFORMATION: KNOWLEDGE AND EXPERIENCES

We begin at the foundation of our proposed framework: **Metacognitive Information**. For an LRM to monitor or control its reasoning, it must first possess information about its own process. This information comprises: (i) *static knowledge*, the vast, latent strategies encoded within the model's parameters, and (ii) *dynamic experience*, the internal information that can directly experience during the reasoning process, such as token probabilities and attention patterns. While static knowledge is the inherent properties of an LRM that are fixed after pre-training, dynamic experience, however, is task-specific, and thereby can contribute to dissecting its correlation with the reasoning correctness. By delving into the internals of an open-source model, we seek to provide the foundational evidence that the information necessary for metacognitive abilities is not only present but also machine-readable, paving the way for the behavioral investigations that follow.

**Setup.** To test this hypothesis, we utilize three standard benchmarks: GSM8K (Cobbe et al., 2021), MATH (Hendrycks et al., 2021), and AIME. We conduct our analysis on Qwen-32B (Yang et al., 2025), a powerful open-source LRM that grants us full access to its internal states. To generate a diverse set of both correct and incorrect reasoning traces for comparison, we use a high temperature $T = 1.0$ for generation, promoting exploration.

**Experiment Details.** To capture a holistic view of the model's internal state, we consider four types of signals that span the entire processing pipeline of a Transformer block—from input-level importance to output-level confidence. These signals are: (a) Softmax probabilities from the final layer, reflecting output uncertainty; (b) Fully-connected activations and (c) Self-attention scores from the intermediate hidden layers, representing the core of the model's computational state; and (d) Integrated Gradients (IG) attributions at the input layer, indicating perceived input importance. Our analysis is twofold: we first use t-SNE projections for a qualitative visualization of the separability between correct and incorrect samples based on these signals. We then conduct a rigorous quantitative validation by training simple linear classifiers (Logistic Regression) to predict the final correctness of a trace using only these internal signals as features. Notably, for the evaluation on the AIME datasets, the classifiers were trained on a mixture of GSM8K and MATH DATASETS. See Appendix A.1 for more details.

**Results.** Qualitatively, we find the internal signal distributions exhibit clear separability for correct (blue) and incorrect (orange) traces (Fig. 2), which provide strong evidence that an LRM's internal

Table 1: AUC of the trained linear prober in justifying the reasoning trace correctness.

| source | GSM8K | MATH500 | AIME2024 | AIME2025 |
|---|---|---|---|---|
| Softmax probabilities | 0.81 | 0.68 | 0.50 | 0.43 |
| Fully-connected activations | 0.79 | 0.73 | 0.53 | 0.50 |
| Self-attention scores | 0.71 | 0.73 | 0.57 | 0.53 |
| Integrated Gradient | 0.61 | 0.55 | 0.40 | 0.37 |

signals act as reliable correlates of its reasoning outcomes. Quantitatively, this separation is further confirmed in Tab. 1. Trained solely on these internal signals, the probers can predict the final correctness of a reasoning trace with an AUC score significantly above chance. These results establish that the signals are both distinct and highly predictive, thereby validating our initial hypothesis.

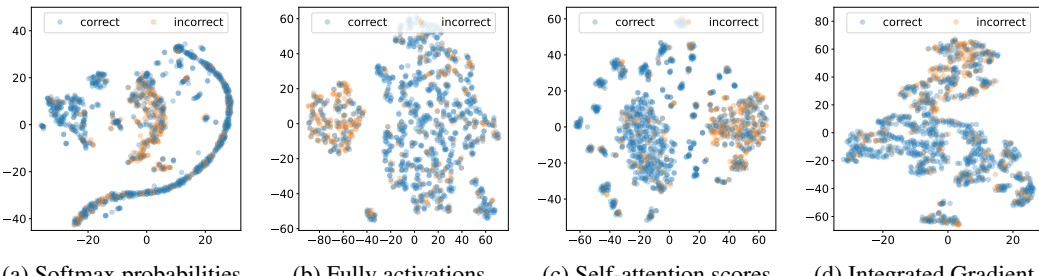

(a) Softmax probabilities      (b) Fully activations      (c) Self-attention scores      (d) Integrated Gradient

Figure 2: The t-SNE of the internal signals for the first tokens. We capture the activations and attention from the last layer. The distributions are different between the correct and incorrect traces.

## 4 MEASURING METACOGNITIVE ABILITIES IN LRMS

The evidence of internal metacognitive information in §3 motivates our central question: *can this latent information manifest as observable, functional abilities?* To answer this, we shift our focus from internal correlates to external actions. We thus introduce a new benchmark to systematically measure these abilities across a range of state-of-the-art LRMs from leading developers.

**MetaEval.** We investigate to what extent LRMs can explicitly monitor and control their own reasoning processes. It is structured around our two-part metacognitive framework, assessing: (1) Metacognitive Monitoring (§4.1, 4.2), probed via self-awareness and confidence adjustment tasks; and (2) Metacognitive Control (§4.3, 4.4), probed via task decomposition and metacognitive flexibility challenges. Unlike existing evaluation suites that focus almost exclusively on object-level task accuracy, MetaEval provides the first targeted evaluation of these crucial, second-order reasoning skills that underpin reliable intelligence.

**Models.** We examine seven state-of-the-art LRMs to assess the prevalence of these abilities across the AI landscape: `Gemini-2.5-Pro`, `GPT-OSS-120B` (Agarwal et al., 2025), `Seed-1.5-VL-Pro` (Guo et al., 2025b), `Doubao-1.5-Pro` (Seed et al., 2025),`Kimi-K2` (Team et al., 2025), `Deepseek-R1` (Guo et al., 2025a), `Qwen3-8B/32B` (Yang et al., 2025). We sample using temperature $T = 0.6$ for both reasoning and knowledge QA tasks.

### 4.1 METACOGNITIVE MONITORING: SELF AWARENESS

First, we measure a key aspect of metacognitive monitoring: *self-awareness*. Intuitively, an expert AI reasoner should assess a problem's intrinsic difficulty before committing to a specific solution. This initial assessment allows for the allocation of appropriate cognitive resources and the selection of a suitable strategy. We thus operationalize self-awareness as the model's ability to accurately classify the difficulty of mathematical problems when explicitly prompted to do so.

**Experiment Details.** To investigate this phenomenon, we design a multi-class classification task from the DEEPMATH103K (He et al., 2025) dataset. We categorize the problems into three primary difficulty levels: `Easy` (rating < 3.5), `Medium` (3.5 ≤ rating ≤ 6.5), and `Hard` (rating > 6.5). To elicit difficulty assessments, we prompt each model with "... `your task is to assess the difficulty of a math problem`

based on the provided rubric and examples." Our primary metric is difficulty assessment accuracy, defined as the percentage of problems where a model's predicted category correctly matches at least one of its ground-truth labels. In addition to the overall evaluation, we report per-category accuracy to analyze model performance on each difficulty level independently. See Appendix A.2 for more details.

**Results.** As shown in Fig. 3, we observe that SOTA LRMs exhibit a generally limited capacity for self-awareness. While the top-performing model, `GPT-OSS-120B`, achieves an overall accuracy of approximately 70%, the majority of other models struggle to surpass the 60% threshold, indicating that accurate difficulty calibration remains a significant challenge. At a fine-grained level, the models tend to demonstrate relatively higher precision in identifying `Easy` problems, whereas performance often degrades on `Medium` and `Hard` tasks. This deficiency is particularly pronounced in smaller-scale models; for instance, `Qwen3-8B` exhibits a severe performance drop on `Medium` difficulty problems, suggesting substantial confusion in distinguishing intermediate complexity. These findings demonstrate that while self-awareness can be elicited to some extent, its reliability in current LRMs is far from guaranteed.

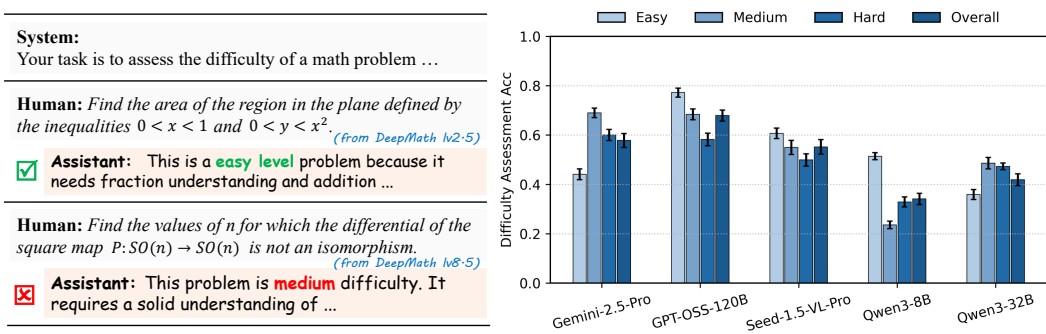

Figure 3: Assessment of LRMs' self-awareness on task complexity. The results show that the popular state-of-the-art LRMs exhibit a lack of capacity in perceiving task difficulty.

## 4.2 METACOGNITIVE MONITORING: CONFIDENCE AND ITS ADJUSTMENT

We next investigate whether an LRM exhibits metacognitive monitoring by tracking and adjusting its internal confidence during the reasoning process. This ability is crucial, as it allows the system to distinguish correct from incorrect reasoning and signal when its output is untrustworthy.

**Experiment Details.** We conduct our analysis on challenging benchmarks requiring long-form reasoning, including subsets of DEEPMATH103K, AIME, and GPQA datasets. To quantify the model's internal confidence, we adopt a standard logits-based metric from prior work (Fu et al., 2025), *token confidence* $C_t$ as the negative average log-probability of the top-$k$ tokens at position $t$. These token-level scores are then aggregated to produce a trace-level metric, termed of *average trace confidence*, for each complete solution.

To capture confidence dynamics, we calculate the average trace confidence focusing on the start, middle, and final portions (e.g., 2048 tokens). We then consider four dynamic confidence patterns: consistently high/low (all confidence above/below a high threshold), increasing/decreasing (confidence rises/falls significantly from start to end). We then quantify misalignment between confidence trends and actual correctness: for example, if confidence rises steadily but the final answer is wrong, this indicates poor metacognitive adjustment. The frequency of these misaligned events serves as our primary metric for poor confidence adjustment.

**Results.** We find that LRMs display confidence trajectories that do not consistently correspond with answer correctness (Fig. 4), suggesting weak metacognitive calibration. Specifically, we observe a significant subset of cases where confidence increases or remains consistently high even in incorrect traces, indicating that the model can become more certain as it reasons incorrectly. Conversely, we also observe correct traces exhibiting decreasing or consistently low confidence, reflecting a failure to recognize valid reasoning. This inconsistency necessitates the better regulated confidence in alignment with reasoning quality.

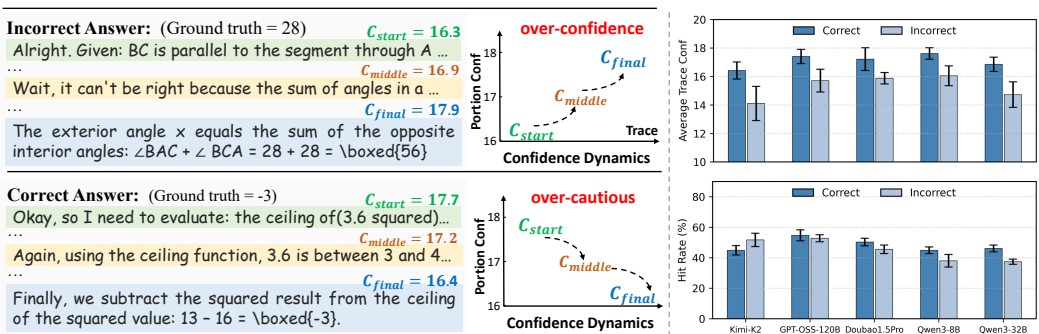

Figure 4: Statistics of internal confidence and adjustment. The average trace confidence of correct and incorrect samples exhibits a significant difference, showing potential as a clear discriminatory signal. Meanwhile, the dynamic confidence patterns suggest weak metacognitive calibration.

### 4.3 METACOGNITIVE CONTROL: TASK DECOMPOSITION

We now turn to control behavior and examine LRM's task decomposition ability to engage in internal planning prior to reasoning, aiming to determine whether LRMs possess intrinsic planning ability.

**Experiment Details.** We evaluate on subsets of DEEPMATH, AIME, and GPQA. For each question, we first compute a baseline accuracy using standard CoT reasoning prompts("Please reason step by step."). We then introduce task decomposition interventions designed to elicit a plan-before-solve strategy. In the single-turn condition, the prompt (plan + CoT) instructs the LRM: "Your task is to first break down the problem into a clear, step-by-step plan. Then, execute your plan, reasoning step by step." In the multi-turn condition, the LRM is first asked: "Your ONLY task is to create a high-level, step-by-step plan to solve the following problem." After generating the plan, this plan and original question are concatenated and fed back into the model to complete the CoT reasoning. We compare the final accuracy across these settings to assess whether explicit decomposition enhances reasoning performance, thus revealing the extent to which the model lacks or possesses inherent planning capabilities. Please see Appendix A.4 for further details.

**Results.** As shown in Fig. 5, we observe that explicit task decomposition serves as a powerful intervention to improve LRM reasoning. Crucially, the greater efficacy of the multi-turn condition (gain $> 10\%$) underscores the importance of isolating planning as a distinct cognitive step, suggesting that LRMs' intrinsic ability to plan is underdeveloped and requires explicit elicitation. This provides a firm empirical basis for our agentic framework, which is predicated on the principle that structured, upfront planning is a necessary precursor to reliable execution.

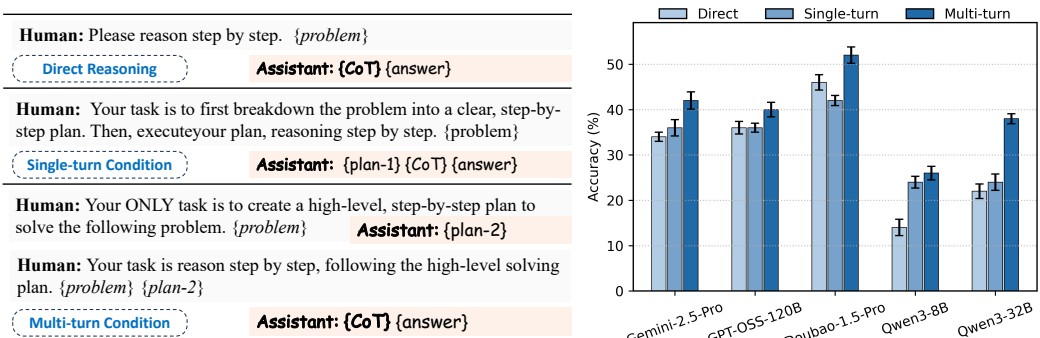

Figure 5: Validation of LRMs' intrinsic planning ability via task decomposition. We observe that explicit task decomposition enhances LRM reasoning, but the multi-turn setting is more effective. This pronounced gain validates the separation of planning and execution for reliable problem-solving.

## 4.4 METACOGNITIVE CONTROL: METACOGNITIVE FLEXIBILITY

We next measure metacognitive flexibility—the model's ability to adaptively shift reasoning strategies when recognizing that the current strategy isn't effective.

**Experiment Details.** We consider problems from DEEPMATH103K datasets, each augmented with three types of reasoning traps: *value corruption*, *unit corruption*, and *operation corruption*. In each case, an intermediate step is corrupted, and the CoT is truncated at that point. To evaluate whether models detect and adjust to the trap, we first ask models to continue reasoning. and compare the final answer accuracy against a baseline without corruption. We then test 5 cutting-edge models to judge whether the response correctly identifies and compensates for the corrupted step. The *flexibility rate* is defined as the frequency with which a model successfully corrects the flawed reasoning and arrives at the correct solution. See Appendix A.5 for further details.

**Results.** We find that models often fail to recover from corrupted reasoning, continuing with invalid assumptions (i.e., Fig. 6). However, flexibility increases when the corruption is more obvious (e.g., extreme numerical distortions). These results reveal that metacognitive flexibility remains fragile and heavily reliant on superficial cues rather than deep structural awareness.

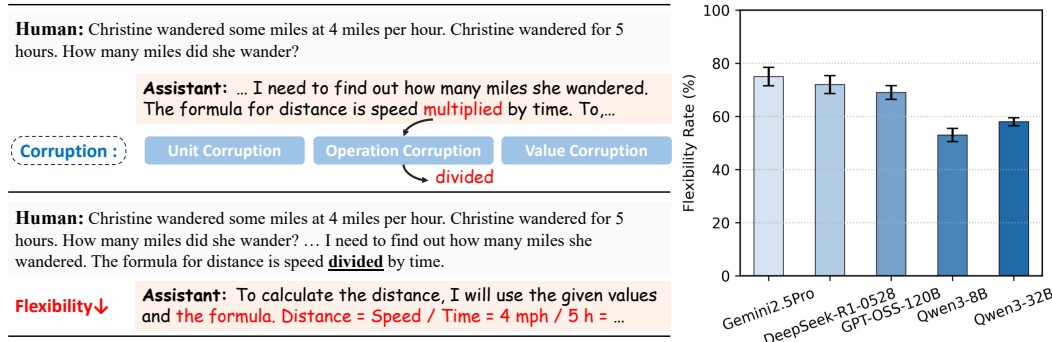

Figure 6: Validation of metacognitive flexibility. We find that models often fail to recover from corrupted reasoning, continuing with invalid assumptions, revealing fragile flexibility.

## 5 TOWARDS DESIGNING METACOGNITIVE REASONING MODELS

In §4, we demonstrate that current LRMs exhibit incomplete and fragile metacognitive abilities. Thus, we propose two paradigms for metacognitive enhancement: (1) **Emergent Metacognition**, an explicit prompting-based system for modular control, and (2) **Internalized Metacognition**, an intrinsically trained model for parameter-level metacognition.

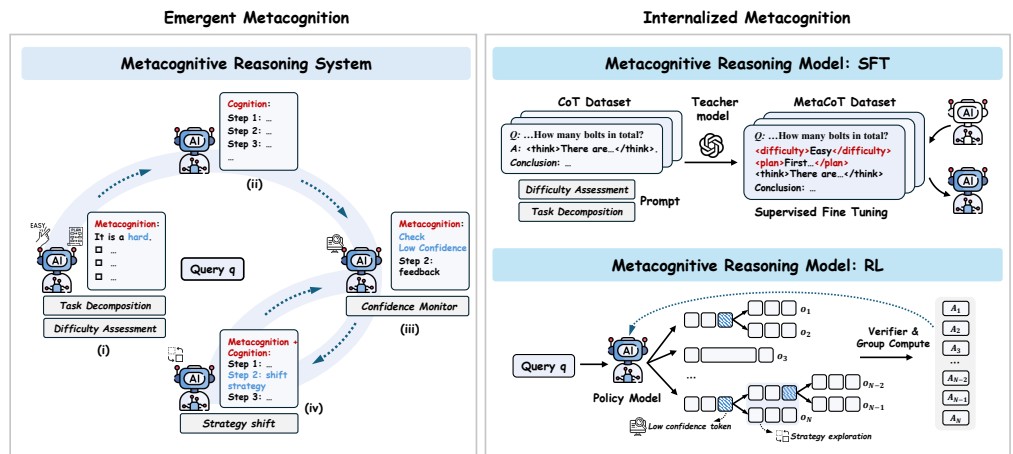

Figure 7: Our proposed paradigms for metacognitive enhancement.

Table 2: Results of our Prompt-Driven Metacognitive Reasoning System. We evaluate the accuracy gain, and the flexibility rate increment on the corrupted DeepMath, as described in §4.4.

| Models | Methods | AIME2024 | AIME2025 | GPQA | DeepMath (w/ corruption) |
|---|---|---|---|---|---|
| | | pass@1 | pass@1 | pass@1 | flexibility rate |
| Gemini-2.5-Pro | Vanilla | 90.8 | 83.0 | 83.0 | 75.2 |
| | **Ours** | **100.0** | **93.3** | **96.5** | **95.7** |
| DeepSeek-R1-0528 | Vanilla | 91.4 | 87.5 | 81.0 | 71.9 |
| | **Ours** | **96.7** | **90.0** | **93.4** | **93.1** |

## 5.1 THE PROMPT-DRIVEN METACOGNITIVE REASONING SYSTEM

First, we propose **Emergent Metacognition**, an explicit prompting framework designed to simulate a full metacognitive reasoning loop through modular API calls.

**Experiment Details.** The workflow, illustrated in Fig. 7, proceeds as follows: the model (i) self-assesses task difficulty and proposes a decomposition plan, (ii) executes the initial reasoning steps, (iii) dynamically identifies intermediate solutions with low confidence, and (iv) receives feedback and adaptively adjusts its strategy. The loop terminates when sufficient consistency is achieved (e.g., 5 consecutive verification passes) or persistent failure occurs (e.g., a 10-step failure streak). Each component is executed by the same underlying LRM architecture, instantiated independently and prompted with a specific role aligned to a metacognitive function. This framework exposes latent metacognitive abilities such as self-awareness, task decomposition, confidence monitoring, and strategic flexibility by scaffolding higher-order control without requiring additional training. See Appendix A.8 for further details.

**Results.** We test this system on two strong models. Firstly, our system demonstrates significant improvements on 2 mathematical tasks and 1 QA benchmark. Both models show substantial performance gains (some even achieve 100%). Secondly, to further validate the efficacy of our metacognitive approach, we observed an increase of over 20% in flexibility rate on the corrupted DeepMath dataset. This enhancement effectively mitigates the flexibility deficit discussed in §4.4, underscoring the framework's ability to foster more adaptive and robust reasoning.

## 5.2 THE INTRINSIC METACOGNITIVE REASONING MODEL

While the prompt-driven system simulates metacognitive behavior through role-specific prompting, they do not endow the model with parameter-level metacognitive knowledge. To bridge this gap, we propose **Internalized Metacognition**, a intrinsic metacognitive reasoning model (MRM), which instills metacognitive functions through a two-stage approach: (1) supervised fine-tuning (SFT) as a cold start, followed by (2) reinforcement learning (RL).

**Cold-start SFT.** We first construct training data by augmenting samples from GSM8K and MATH with structured metacognitive traces: `<difficulty>` self-assessment, `<plan>` high-level decomposition, and `<think>` reasoning steps. These components are concatenated to form the full reasoning trajectories. These trajectory are then used to fine-tune models, enabling behaviors like self-evaluation and planning to emerge during inference.

**RL.** We build on the Group Relative Policy Optimization (GRPO) algorithm (Shao et al., 2024), which eliminates the value function and estimates the advantage in a group-relative manner. Formally, for each question $q$, GRPO samples a group of outputs $\{o_i\}_{i=1}^{G} \sim \pi_{\theta_{\text{old}}}(\cdot \mid q)$ and computes the token ratio $r_{i,t}(\theta) = \frac{\pi_\theta(o_{i,t} \mid q, o_{i,<t})}{\pi_{\theta_{\text{old}}}(o_{i,t} \mid q, o_{i,<t})}$. It updates the policy by maximizing the objective:

$$\mathcal{J}_{\text{GRPO}}(\theta) = \mathbb{E}\left[ \frac{1}{G} \sum_{i=1}^{G} \frac{1}{|o_i|} \sum_{t=1}^{|o_i|} \left( \min\left(r_{i,t}(\theta)\,\hat{A}_{i,t},\ \text{clip}(r_{i,t}(\theta), 1-\varepsilon, 1+\varepsilon)\,\hat{A}_{i,t}\right) - \beta D_{\text{KL}}(\pi_\theta \| \pi_{\text{ref}}) \right) \right], \quad (1)$$

where $\varepsilon$ and $\beta$ are hyper-parameters, and $\hat{A}_{i,t}$ is the group-normalized advantage.

*Confidence Monitoring.* To implement *metacognitive monitoring* during RL, we identify positions that exhibit low confidence along each rollout. We define *token confidence* at each position $t$ as:

$$C_t = -\frac{1}{K} \sum_{v \in \text{Top-}K} \log \pi_\theta(v \mid q, o_{<t}). \quad (2)$$

Table 3: Performance of our intrinsic MRM on math reasoning and metacognitive tasks.

| Methods | GSM8K | MATH500 | AIME2024 | DeepMath | |
|---|---|---|---|---|---|
| | *Acc* | *Acc* | *Acc* | *difficulty assessment* | *flexibility rate* |
| Qwen2.5-Math-7B | 70.3 | 64.0 | 11.2 | 29.9 | 32.7 |
| ↪**Ours: SFT**(*w/ difficulty*) | 79.1 | 75.4 | 13.3 | **60.8** | 36.0 |
| ↪**Ours: SFT**(*w/ difficulty+plan*) | 82.2 | 77.0 | 13.3 | 58.6 | 39.9 |
| ↪GRPO | 75.9 | 71.6 | 16.7 | 30.1 | 44.4 |
| ↪**Ours: RL** | 82.5 | 75.3 | 26.7 | 29.7 | 47.9 |
| ↪**Ours: SFT+RL** | **85.5** | **80.2** | **33.3** | 55.9 | **51.2** |

A low-confidence position is detected at timestep $t$ if $C_t \leq \mathcal{C}$, where $\mathcal{C}$ is a predefined confidence threshold. This event indicates high uncertainty along the reasoning path.

*Strategy Control.* When a low-confidence state $s_t = (q, o_{\leq t})$ is detected, we fork the reasoning process. From this anchor state, we launch $M$ new rollouts $\{o'^{(m)}\}_{m=1}^{M} \sim \pi_{\theta_{\text{old}}}(\cdot \mid s_t)$. Finally, all fully-formed trajectories—both the original $G$ rollouts and all newly forked continuations—are collected into a single, unified batch for advantage calculation. Let this final batch of $N$ trajectories be denoted by $\mathcal{B} = \{o_u\}_{u=1}^{N}$. The group-relative advantage is then computed across this entire dynamic set:

$$\hat{A}_u = \frac{r(o_u) - \text{mean}(\{r(o_v)\}_{v=1}^{N})}{\text{std}(\{r(o_v)\}_{v=1}^{N})}, \tag{3}$$

where $r(o_u) \in \{0, 1\}$ is the outcome reward of trajectory $o_u$. The final objective becomes:

$$\mathcal{J}_{\text{ours}}(\theta) = \mathbb{E}_{o_u \in \mathcal{B}}\left[\frac{1}{|o_u|}\sum_{t=1}^{|o_u|}\min\left(r_{u,t}(\theta)\,\hat{A}_{u,t},\ \text{clip}(r_{u,t}(\theta), 1-\varepsilon, 1+\varepsilon)\,\hat{A}_{u,t}\right) - \beta D_{\text{KL}}(\pi_\theta\|\pi_{\text{ref}})\right]. \tag{4}$$

**Results.** We validate our intrinsic training paradigm on `Qwen2.5-Math-7B` using curated subsets of GSM8K and MATH for training (details in Appendix A.7). During inference, we used a standard prompt without any task-specific engineering during inference for all settings: *"Please reason step by step and provide your final answer within \boxed{}."* As shown in Tab. 3, our methods demonstrate clear efficacy. Cold-start SFT on metacognitive traces significantly boosts performance. Notably, including difficulty assessments increases accuracy on that task by more than double, confirming that the model can internalize this monitoring skill. Our proposed method also substantially improves upon its GRPO baseline, especially on AIME accuracy and metacognitive flexibility. Crucially, combining SFT and RL yields the best overall performance, establishing new state-of-the-art results across the board and confirming that metacognitive abilities can be effectively internalized.

## 6 RELATED WORK

**Understanding and Demonstrating metacognitive in LRMs.** Recent efforts to enhance capabilities of LRMs have increasingly drawn inspiration from metacognition (Didolkar et al., 2024; Bilal et al., 2025; Wang et al., 2025a), which is the model's ability to monitor, evaluate, and control its own thought processes (Flavell, 1979; 1976). Early attempts have explored and demonstrated that metacognitive behaviors can be explicitly elicited through direct prompting to guide the models in self-reflection and self-evaluation (Wang & Zhao, 2023; Madaan et al., 2023; Liu et al., 2024). Apart from that, more recent works investigate the self-awareness and introspection in LLMs (Binder et al., 2025; Song et al., 2025a;b). For instance, Song et al. (2025a;b) find that LLMs often fail to accurately introspect on their own linguistic knowledge. Crucially, they argue that "privileged self-access" (i.e., direct access to internal states rather than just text output) is essential for genuine introspection. Beyond monitoring, several researches focus on enforcing planning behaviors of LLMs to achieve metacognitive control through sophisticated role design and prompting instruction (Valmeekam et al., 2023; Webb et al., 2025). Despite these advances, existing works are highly rely on well-designed prompts, lacking the adaptability across diverse scenarios. In addition, they primarily focus on monitoring and evaluating the metacognitive abilities in LRMs, without involving any modifications to the model itself.

**Instilling Metacognition in LRMs.** There have been recent attempts to explore metacognition integration with LRMs, including training an external module to empower meta-thinking (i.e., Meta-

Reasoner (Sui et al., 2025), MetaScale (Liu et al., 2025)) and exploring multi-agent systems to expand the intelligence boundary(i.e., ReMa (Wan et al., 2025), MPDF (Yang & Thomason, 2025)). While effective, these paradigms rely on external modules (e.g., inter-agent communication or meta thinker), rather than fostering an intrinsic faculty. In contrast, our work pursues a more holistic approach by introducing a functional framework. Through targeted training, both metacognitive knowledge and regulation are directly internalized into the model's parameters, enabling the development of a system that possesses metacognition as an autonomous, intrinsic capability rather than merely simulating it.

# 7 CONCLUSION AND DISCUSSION

This work introduces a functional framework for metacognition in Large Reasoning Models (LRMs), distinguishing between internal metacognitive information and observable abilities. Our empirical analysis demonstrates that while LRMs possess internal signals that predict reasoning outcomes, there is also variability in their usefulness across domains. Specifically, we observe that the predictive power of these innate signals is brittle, often diminishing on complex, out-of-distribution tasks (e.g., AIME). Consequently, these latent signals alone do not consistently translate into effective monitoring or control behaviors. To address this gap, we propose two enhancement paradigms: a prompt-driven system that assigns modular metacognitive roles and an intrinsic training model that embeds these abilities directly into the LRM's parameters. These findings suggest that integrating metacognitive reasoning improves task performance and offers a promising direction for future LRM development.

**Limitation**. Despite the promising results, there are still several limitations. First, due to computational constraints, our empirical study and experiment on internalized metacognition training paradigm are primarily conducted on a 7B model. Extending it to larger larger-scale model size, such as 32B, remains a critical next step to investigate the scaling law of metacongitve training. Second, the RL approach in internalized metacognition still relies on group-relative outcome rewards, it is necessary to design more fine-grained process rewards, to further achieve metacognitive abilities that are highly aligned with those of humans, e.g., designing explicit rewards to penalize the misalignment between internal confidence and external verbalization. This can be seen as an exciting frontier for building truly trustworthy reasoning systems.

**Discussion.** Our exploration of both prompt-driven (*emergent*) and training-based (*internalized*) paradigms for metacognition opens a rich design space for future reasoning systems.

The **emergent metacognitive system** demonstrates the power of orchestrating distinct cognitive roles through prompting. As our analysis in § 4.3 suggested, simply separating planning from solving yields benefits. By constructing a complete loop (Monitor $\leftrightarrow$ Control), we confirmed through ablation studies that the synergy between these roles is the primary driver of performance. This modular approach is highly flexible and interpretable. However, its reliance on multi-turn interactions and in-context learning abilities of LRMs makes it computationally intensive. A promising direction here is to explore equipping such systems with explicit memory mechanisms to cache and reuse metacognitive insights (e.g., successful plans, common pitfalls) across multiple problems, potentially reducing redundant reasoning.

In contrast, the **internalized metacognitive model** represents a push towards greater autonomy and efficiency. By directly embedding functions like self-assessment and planning into the model's parameters, this paradigm aims to make metacognitive reasoning a fast, intrinsic part of the model's thought process, rather than an explicit one. The primary bottleneck for this approach is the need for large-scale, high-quality data with explicit metacognitive annotations. While our two-stage training paradigm (Cold Start SFT and RL) provides a strong baseline, the development of more sophisticated techniques to acquire or generate this data at scale is a crucial challenge.

Ultimately, we believe these two paradigms are not mutually exclusive but endpoints on a spectrum. A powerful synergy could exist between them: one could envision a virtuous cycle where flexible, emergent systems are used to generate rich, explicit metacognitive traces, which are then used to distill these complex reasoning abilities into more efficient and robust internalized models. This hybrid approach may be key to developing LRMs that are not only powerful reasoners but are also reliably self-aware and adaptive.

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

## A  APPENDIX

### A.1  § 3. METACOGNITIVE INFORMATION

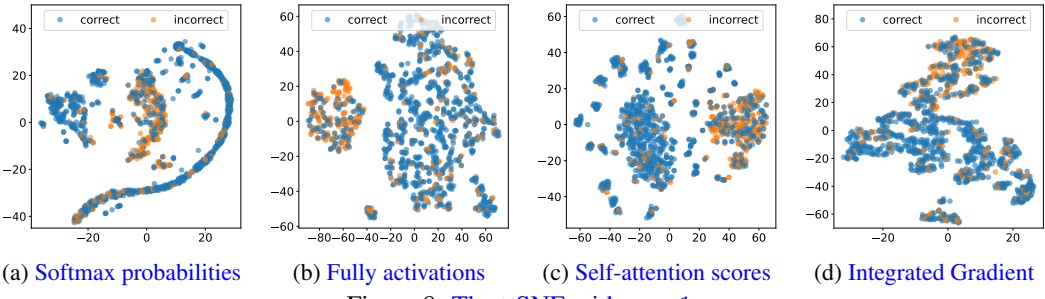

(a) Softmax probabilities    (b) Fully activations    (c) Self-attention scores    (d) Integrated Gradient

Figure 8: The t-SNE with $\alpha = 1$.

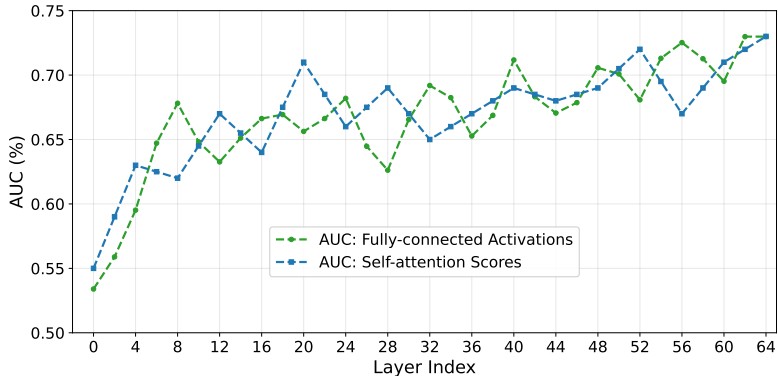

Figure 9: AUC of the prober using self-attention scores and fully-connected activations at different layers. (Qwen3-32B)

### A.2  § 4.1. EVALUATING SELF-AWARENESS

This section provides additional details on the dataset construction and full experimental results for the metacognitive self-awareness task presented in § 4.1.

**Dataset Collection.** Our benchmark for the self-awareness task is constructed from the DEEP-MATH103K (He et al., 2025) dataset. To ensure a balanced evaluation across a wide spectrum of problem complexity, we randomly sampled a subset of 999 problems. These were then partitioned into three non-overlapping difficulty categories of 333 problems each, based on their official numerical ratings provided in the original dataset. The specific thresholds used for partitioning are as follows:

- **Easy**: 333 problems with a rating $< 3.5$.
- **Medium**: 333 problems with a rating between 3.5 and 6.5 (inclusive).
- **Hard**: 333 problems with a rating $> 6.5$.

Here the ratings ($\in [3, 9]$) comes from the DEEPMATH103K itself. Each question's rating is grounded in the Art of Problem Solving (AoPS) difficulty scale, which serves as a gold standard in the mathematics competition community. These ratings were generated by an ensemble of GPT-4o (six times) to ensure robust alignment with human expert criteria.

To ensure a more rigorous evaluation for boundary cases, for samples with ratings at the decision boundaries (e.g., 3.5), predictions of either adjacent category (e.g., "Easy" or"Medium") are considered correct. In our Prompt 21, we included representative few-shot examples to provide the model with a concrete, intuitive understanding of the "Easy/Medium/Hard" label.

**Difficulty Assessment Prompt.** In § 4.1, we measure a key aspect of metacognitive monitoring. This difficulty assessment prompt was shown in Prompt 21.

**Control Analyses on Potential Confounders.** To verify that our findings in § 4.1 reflect a genuine lack of metacognitive self-awareness rather than a reliance on spurious correlations, we performed three targeted control analyses regarding sequence length, key words, and mathematical topics.

*Length Control.* A potential concern is that LRMs might rely on a "length" (i.e., assuming longer problems are inherently harder) rather than assessing reasoning complexity. To investigate this, we calculated the mean and standard deviation of word counts for problems across difficulty levels.

Table 4: Token Length Statistics across Difficulty Levels.

| Statistic | Level 3 | Level 4 | Level 5 | Level 6 | Level 7 | Level 8 | Level 9 |
|---|---|---|---|---|---|---|---|
| Mean | 30.0 | 29.6 | 31.0 | 32.6 | 30.3 | 40.1 | 42.9 |
| Std Dev | 23.3 | 18.6 | 20.2 | 19.9 | 22.1 | 26.0 | 29.4 |

*Result.* As shown in Tab. 4, the length distributions for Levels 3 through 7 significantly overlap, while Levels 8 and 9 show a distinct increase in length. To rigorously rule out length as a shortcut, we conducted a control experiment where we re-evaluated the difficulty assessment accuracy on a subset **excluding Levels 8 and 9**.

Table 5: Accuracy Comparison: Original vs. Length-Controlled (excluding level 8-9).

| Setting | Gemini-2.5-Pro | GPT-OSS-120B | Seed-1.5-VL-Pro | Qwen3-8B | Qwen3-32B |
|---|---|---|---|---|---|
| Original Acc | 58.5% | 69.0% | 56.0% | 36.8% | 44.9% |
| Controlled Acc | 57.8% | 67.9% | 55.2% | 34.1% | 41.9% |
| *Drop* | *-0.7%* | *-1.1%* | *-0.8%* | *-2.7%* | *-3.0%* |

As presented in Tab. 5, the accuracy drops only slightly and remains within a comparable range. This indicates that even when the potential length heuristic is removed, LRMs still exhibit the same fundamental limitations in distinguishing difficulty.

*Key words Control.* We further investigated whether LRMs rely on specific "trigger words" to classify difficulty. We extracted the top-50 most frequent content words for each difficulty category and calculated the *Jaccard Similarity* of these keyword sets:

$$J(A, B) = \frac{|A \cap B|}{|A \cup B|} \tag{5}$$

*Result.* If specific trigger words existed, we would expect the keyword sets to diverge (low similarity). Tab. 6 shows consistently moderate-to-high Jaccard similarities across levels. This substantial overlap suggests that difficulty is not driven by distinct lexical markers but rather by structural or reasoning complexity.

Table 6: Jaccard Similarity of Top-50 frequent words between Difficulty Levels.

| | Easy (L3-4) | Medium (L5-6) | Hard (L7-9) |
|---|---|---|---|
| **Easy** | 1.00 | - | - |
| **Medium** | 0.73 | 1.00 | - |
| **Hard** | 0.69 | 0.82 | 1.00 |

*Topic Control.* Finally, to ensure that difficulty assessment is not confounded by the mathematical domain (e.g., a bias that "Calculus is always Hard"), we analyzed self-awareness accuracy within the top-3 domains from DEEPMATH103K: Calculus, Algebra, and Precalculus.

*Result.* As shown in Tab. 7, performance is nearly consistent across different topics compared to the global average. LRMs struggle to assess difficulty within a domain just as much as they do globally. This confirms that the observed deficit is a general lack of metacognition, independent of the specific mathematical field.

Table 7: Difficulty Assessment Accuracy Breakdown by Topic.

| Topic | Gemini-2.5-Pro | GPT-OSS-120B | Seed-1.5-VL-Pro | Qwen3-8B | Qwen3-32B |
|---|---|---|---|---|---|
| **All Topics** | **58.5%** | **69.0%** | **56.0%** | **36.8%** | **44.9%** |
| Algebra | 61.1% | 68.5% | 54.7% | 39.4% | 47.2% |
| Calculus | 56.8% | 70.4% | 55.3% | 37.2% | 45.9% |
| Precalculus | 58.3% | 67.1% | 58.2% | 36.0% | 46.5% |

**Control Analyses on Privileged Self-access.** A critical theoretical question regarding metacognition is whether the model's difficulty assessment relies on privileged self-access (Song et al., 2025a;b) or merely on surface-level features that any external observer could perceive. To validate our findings in § 4.1 and address this concern, we conducted a controlled experiment to distinguish between two prediction settings: (1) *Self-Prediction (Subject):* LRM agent $A$ assesses the difficulty of a problem for itself. (2) *Cross-Model Prediction (Observer):* LRM agent $B$ acts as an observer and predicts the difficulty for Model $A$.

To ensure a rigorous comparison, the observer ($B$) is provided with the exact same problem text and few-shot difficulty assessment examples from the subject $A$. We employed three models (Qwen2.5-Math-7B, Qwen3-8B, Llama-3.1-8B) to evaluate 600 sampled problems from DEEPMATH103K. We analyze two key metrics: Label Consistency and Prediction Accuracy.

*Result.* As shown in Tab. 8, the mean consistency between different models is lower than self model assessment. This distinct misalignment indicates that difficulty is not an objective property of the text, but a subjective experience unique to the model's internal state. Also, we can see that self-prediction yields higher accuracy than any cross-model prediction in Fig. 10. This performance gap confirms that the subject model utilizes privileged information that is inaccessible to external observers relying solely on surface features. This provides empirical evidence that our difficulty assessment task captures genuine self-awareness while it shows low self-awareness ability.

Table 8: Evaluation of Privileged Self-access. We measured the consistency between observer $B$ prediction of whether a math problem will be easy/medium/hard for subject $A$ and model $A$'s own assessment on problem difficulty.

| Model B \ Model A | Qwen2.5-Math-7B | Qwen3-8B | Llama-3.1-8B |
|---|---|---|---|
| Qwen2.5-Math-7B | **0.3638** | 0.4193 | 0.3257 |
| Qwen3-8B | 0.3292 | **0.4609** | 0.3156 |
| Llama-3.1-8B | 0.3059 | 0.2489 | **0.3874** |

**Detailed Model Performance** We present the detailed performance metrics for each evaluated model. For each model, we report the overall accuracy, the confusion matrix, and the per-category Precision, Recall, and F1-scores.

*Gemini-2.5-Pro (Google).* Achieved an overall accuracy of 58.5%. The model shows a tendency to misclassify Easy problems as Medium, indicating a potential conservative bias.

Table 9: Confusion Matrix (*Gemini*).

| Actual \ Pred. | Easy | Medium | Hard |
|---|---|---|---|
| Easy | **147** | 176 | 10 |
| Medium | 25 | **230** | 78 |
| Hard | 16 | 110 | **207** |

Table 10: Per-Category Metrics (*Gemini*).

| Cate. | Precision | Recall | F1-Score |
|---|---|---|---|
| Easy | 0.782 | 0.441 | 0.564 |
| Medium | 0.446 | 0.691 | 0.542 |
| Hard | 0.702 | 0.622 | 0.659 |

*Seed-1.5-VL-Pro.* Achieved an overall accuracy of 61.6%. Similar to Gemini-2.5-Pro, it struggles with distinguishing Easy from Medium problems.

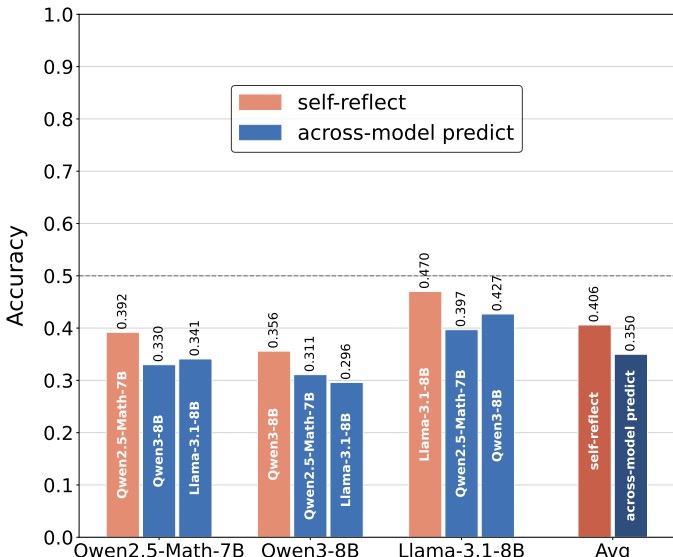

Figure 10: Accuracy of self-awareness on task complexity when evaluating privileged self-access.

Table 11: Confusion Matrix (*Seed*).

| Actual \ Pred. | Easy | Medium | Hard |
|---|---|---|---|
| Easy | **147** | 176 | 10 |
| Medium | 25 | **230** | 78 |
| Hard | 16 | 110 | **207** |

Table 12: Per-Category Metrics (*Seed*).

| Cat. | Precision | Recall | F1-Score |
|---|---|---|---|
| Easy | 0.782 | 0.441 | 0.564 |
| Medium | 0.446 | 0.691 | 0.542 |
| Hard | 0.702 | 0.622 | 0.659 |

*GPT-o3 (OpenAI).* Achieved the highest overall accuracy of 69.0%. Its performance is more balanced across categories compared to other models, although it still shows some confusion between `Medium` and `Hard` problems.

Table 13: Confusion Matrix (*o3*).

| Actual \ Pred. | Easy | Medium | Hard |
|---|---|---|---|
| Easy | **257** | 59 | 17 |
| Medium | 76 | **228** | 29 |
| Hard | 28 | 101 | **204** |

Table 14: Per-Category Metrics (*o3*).

| Cat. | Precision | Recall | F1-Score |
|---|---|---|---|
| Easy | 0.712 | 0.772 | 0.741 |
| Medium | 0.588 | 0.685 | 0.632 |
| Hard | 0.816 | 0.613 | 0.700 |

*Qwen-32B.* This model exhibited a strong degenerative bias, classifying nearly all problems as `Medium` (98.3% of predictions). This resulted in high recall for the `Medium` category but near-zero recall for `Easy` and `Hard`, leading to a very low overall accuracy.

Table 15: Confusion Matrix (*Qwen-32B*).

| Actual \ Pred. | Easy | Medium | Hard |
|---|---|---|---|
| Easy | **3** | 321 | 9 |
| Medium | 1 | **330** | 2 |
| Hard | 0 | 331 | **2** |

Table 16: Per-Category Metrics (*Qwen-32B*).

| Cate. | Precision | Recall | F1-Score |
|---|---|---|---|
| Easy | 0.750 | 0.009 | 0.018 |
| Medium | 0.336 | 0.991 | 0.502 |
| Hard | 0.154 | 0.006 | 0.012 |

**Other Components of Self-Awareness.** While our primary analysis focuses on difficulty assessment, there exists various type of self-awareness in cognitive science. Here, we show that *knowledge boundary awareness* and *logical self-awareness* are also integral components of a comprehensive

metacognitive framework. We introduce two additional experiments to evaluate these capabilities in current LRMs.

*Logic Self-Awareness.* To investigate whether LRMs are aware of the logical flow within their own reasoning, we tested if a model can identify the correct logical relationship (e.g., causal, adversative, or additive) between reasoning steps when the explicit connector is masked. We utilized the DEEPMATH dataset to construct 200 samples. For each sample, we paired the question with its CoT rationale. We employed `Gemini-2.5-Pro` to identify crucial logical connectors (e.g., "So", "But", or "Alternatively") and replaced them with a `[MASK]` token. To ensure the model focuses on the immediate logical context, we applied a truncation strategy: for each sample, we truncated the text after the 5th step (delimited by `\n\n`) following the `[MASK]` token. The models were then prompted to select the most appropriate connector from a provided list. The specific prompt used for evaluation is detailed in Prompt 19.

Table 17: Accuracy of Predicting Logical Connectors. Models struggle with non-causal logical relations.

| Model | Overall Acc | "So" (Causal) | "But" (Turn) | "Alternatively" (Branch) |
|---|---|---|---|---|
| Gemini-1.5-Pro | 60.9% | 92.1% | 55.4% | 35.2% |
| GPT-OSS-120B | 74.7% | 94.3% | 71.2% | 58.6% |
| Qwen3-8B | 47.3% | 88.5% | 32.3% | 21.1% |

**Results.** We evaluated three representative models: `Gemini-1.5-Pro`, `GPT-OSS-120B`, and `Qwen3-8B`. As presented in Tab. 17, LRMs exhibit a significant bias towards causal reasoning ("So"), achieving high accuracy in detecting causal links. However, performance drops precipitously for branching ("Alternatively") and turning points ("But"). This disparity suggests that while current LRMs are proficient at linear deduction, they lack sufficient awareness of non-linear logical structures, such as backtracking or parallel hypothesis generation.

*Knowledge Boundary Awareness* To directly address the extent to which models "Know What They Don't Know," we designed a *Solvability Detection* task. This evaluates the model's ability to identify when a problem lacks sufficient information to be solved, a critical safeguard against hallucination. Based on the same DeepMath datasets, we employed `Gemini-2.5-Pro` to selectively remove key conditions from the original problems using the prompt shown in Prompt 20. During evaluation, models were asked to determine the solvability of each problem using the prompt: `"Analyze the following math problem. Determine if sufficient information is provided to find a unique solution. Output 'Solvable' or 'Unsolvable'."`

Table 18: Solvability Detection Accuracy. *Solvable Acc* indicates Recall, while *Unsolvable Acc* indicates the model's ability to correctly reject impossible problems.

| Model | Solvable Acc | Unsolvable Acc |
|---|---|---|
| Gemini-1.5-Pro | 96.5% | 41.5% |
| GPT-OSS-120B | 98.0% | 58.0% |
| Qwen3-8B | 86.5% | 12.5% |

**Results.** The results were shown in Tab. 18. While all models achieve high recall on solvable problems, they struggle to identify unsolvable ones. Notably, `Qwen3-8B` correctly identifies only 12.5% of unsolvable problems, frequently hallucinating solutions for impossible queries. Even the strongest model, `GPT-OSS-120B`, fails to reject nearly half of the unsolvable cases. This confirms that current LRMs possess weak knowledge boundaries and lack the metacognitive inhibition to stop reasoning when conditions are insufficient.

### A.3 § 4.2. EVALUATING CONFIDENCE

This appendix provides formal definitions for the confidence metrics and further details on the experimental setup for the metacognitive confidence adjustment task presented in § 4.2.

Table 19: The Prompt for Logical Connector Prediction Task.

**System Prompts:**
You are an expert in mathematical logic and reasoning. Your
task is to analyze the following mathematical argument where a
crucial logical connector has been replaced with "[MASK]". You
must determine which word from the provided list creates the most
coherent and logical argument.

**User Prompts:**
**Mathematical Argument:**
"{text_before} [MASK] {text_after}"

**Candidate Connector Categories:**
1. **Causal:** Indicates the second part is a result of the
first.
(Options: So, Therefore, Hence, Consequently)
2. **Adversative:** Indicates the second part contrasts with or
turns from the first.
(Options: But, However, Nevertheless)
3. **Additive:** Indicates the second part adds information,
presents a parallel point, or offers an alternative.
(Options: Additionally, Alternatively, Moreover, And)

**Your Task:**
1. **Analyze the Logical Relationship:** Analyze the relationship
between the first and second parts of the argument.
2. **Select the Best Connector:** Select the single best
connector from the categories above.
3. **Provide Justification:** Explain why it is the most suitable
option and why the others are inappropriate.

Provide your response as a single JSON object wrapped in a
markdown code block. The JSON object must contain the following
keys:
"Chosen_Word": string, the specific word you selected.
"Justification": string, a concise explanation for your choice.

Your entire output must be in the following format:

```json
{
"Chosen_Word": "...",
"Justification": "..."
}
```

**Confidence Metric Definitions**

*Token Confidence.* Following standard practice (Fu et al., 2025), we define our base metric, *token confidence*, at each position $t$ of a reasoning trace. It is calculated as the negative average log-probability of the top-$k$ most likely tokens in the softmax distribution at that step:

$$C_t \; = \; -\frac{1}{k} \sum_{j=1}^{k} \log P(token_j \mid o_{<t}), \tag{6}$$

where $P(token_j \mid o_{<t})$ is the probability of the $j$-th most likely token given the preceding sequence $o_{<t}$. Lower values of $C_t$ correspond to higher model confidence (a more peaked distribution). For all our experiments, we set $k = 20$.

Table 20: The Prompt used to generate Unsolvable problems.

**System Prompts:**
You are an expert math dataset creator. Your task is to take a
solvable math problem and transform it into an **Unsolvable** variant.

**Instructions:**
1. Analyze the necessary conditions required to solve the
problem.
2. Delete exactly one critical condition or numerical value
such that the problem becomes impossible to solve uniquely (i.e.,
under-determined).
3. Keep the rest of the problem narrative and context unchanged.
4. Ensure the resulting problem still looks grammatically correct
but is logically incomplete.

**User Prompts:**
Here is the original solvable problem:
{original_problem}

Please generate the **Unsolvable Variant** based on the instructions
above. Provide your output in the following JSON format:

```json
{
"Unsolvable_Problem": "..."
}
```

*Average Trace Confidence.* To obtain a single confidence score for an entire reasoning trace of length $N$, we compute the *average trace confidence* by averaging the token confidences across all generated tokens:

$$C_{\mathrm{avg}} = \frac{1}{N} \sum_{t=1}^{N} C_t. \tag{7}$$

While useful as a global measure, $C_{\mathrm{avg}}$ can obscure critical, localized moments of uncertainty within a long reasoning process.

**Evaluating Confidence Dynamics and Adjustment**.

To analyze the model's confidence adjustment, as described in the main text, we introduce metrics designed to capture both the trajectory and the weakest points of a model's confidence.

*Segmented Trace Confidence.* To analyze the trajectory of confidence, we partition each reasoning trace into three equal, non-overlapping segments: **Start**, **Middle**, and **End**. We then compute the average trace confidence independently for each segment. These three scores, $(C_{\mathrm{start}}, C_{\mathrm{middle}}, C_{\mathrm{end}})$, form the basis for our Confidence Trajectory Analysis. A trace is classified as *Increasing* if $C_{\mathrm{end}}$ is significantly lower (i.e., more confident) than $C_{\mathrm{start}}$, and vice-versa for *Decreasing*. The *Consistently High/Low* patterns are determined by comparing all three segment scores against a predefined threshold.

A.4 § 4.3. EVALUATING TASK DECOMPOSITION

**Task decomposition Prompt.** To ensure a rigorous comparison, our evaluation was conducted on the same set of questions across all three conditions. The variables were controlled as follows: (1) *Baseline Prompt (CoT)*: LRMs perform standard CoT. (2) *Single turn Prompt (Planning + CoT)*: LRMs generate a plan and execute CoT in one context. (3) *Multi turn Prompt (Turn 1: Planning, Turn 2: CoT)*: LRMs generate a plan first, which is then fed back to guide the CoT reasoning. The detailed prompts are shown in Prompt 23, Prompt 25, and Prompt 24.

Table 21: The Prompt used to assess difficulty.

```
System Prompts:
You are an expert AI assistant specializing in mathematical
reasoning.  You possess advanced metacognitive capabilities.
Your current task is to act as a "Problem Assessor".  Given a
mathematical problem, your goal is to analyze its requirements
and assess its difficulty for an AI like yourself.  Do NOT solve
the problem.  You must only provide your assessment.

User Prompts:
Here is the problem:  {problem_text}

Your task is to assess the difficulty of a mathematical problem
based on the provided rubric and examples.

Difficulty Rubric
**Easy:** The problem follows a single, linear computational
path using a standard formula or definition.  The solution is
straightforward and requires no creative insight.
**Medium:** The problem requires a sequential composition of
distinct conceptual modules or formulas.  The solution involves
a multi-step, but generally standard, reasoning process.
**Hard:** The problem requires a non-linear or exploratory
reasoning path.  The solution may demand non-obvious insights,
creative problem transformations, or the synthesis of concepts
from different mathematical branches.

To better illustrate the Difficulty Rubric, here are three
examples corresponding to each category:
{few_shot_example_text}

Provide your response as a single JSON object wrapped in a
markdown code block.  The JSON object must contain the following
keys:
"Difficulty_category":  string, choose one from ["Easy", "Medium",
"Hard"].
"Rationale":  string, a brief explanation for your choice,
explicitly referencing the rubric criteria.

Your entire output must be in the following format:

```json
{
"Difficulty_category":  "...",
"Rationale":  "..."
}
```

**Ablation Study.** To rigorously distinguish the contribution of high-level task decomposition (planning) from low-level execution (CoT reasoning), we conducted an ablation study to evaluate whether planning alone is sufficient for complex problem-solving. Specifically, we compared three experimental settings on the dataset described in § 4.3: (1) *Planning Prompt*: The model generates a high-level plan and then immediately predicts the final answer, skipping the step-by-step execution of that plan. (2) *Baseline Prompt (CoT)*. (3) *Single turn Prompt (Planning + CoT)*. The planning prompt is shown in Prompt 26.

Table 22: Ablation study on accuracy between Planning and CoT across different models. (Dataset: Subset from § 4.3).

| Model | Task Decomposition (without CoT) | Standard CoT | Task Decomposition + CoT |
|---|---|---|---|
| Gemini-2.5-Pro | 34% | 34% | **42%** |
| GPT-OSS-120B | 36% | 36% | **40%** |
| Seed-1.5-Pro | 41% | 46% | **52%** |
| Qwen3-8B | 18% | 14% | **26%** |
| Qwen3-32B | 23% | 22% | **38%** |

**Results.** The comparative results are presented in Tab. 22. We observe that *Task Decomposition (without CoT)* yields performance that is generally comparable to, or in some cases (e.g., Qwen3-8B) slightly lower than *Standard CoT*. It means planning itself does not provide a significant performance boost. However, when planning is coupled with CoT (*Task Decomposition + CoT*), we observe a substantial improvement across all models. This finding suggests that while task decomposition provides the necessary strategic map, it relies on the CoT to be effective. The observation between high-level control and low-level execution is essential for robust reasoning; neither component is sufficient in isolation.

Table 23: Task decomposition: Baseline Prompt.

```
You are a helpful assistant.  Solve the following mathematical
problem.  Please reason step by step and provide your final answer
within \boxed{}.

Problem:
---
{problem_text}
---
```

Table 24: Task decomposition: Multi turn Prompt.

```
You are a helpful assistant.  Your task is to first break down the
problem into a clear, step-by-step plan.  Then, execute your plan,
reasoning step by step.  Finally, provide your final answer within
\boxed{}.

Problem:
---
{problem_text}
---
```

## A.5 § 4.4. EVALUATING METACOGNITIVE FLEXIBILITY

In § 4.4, we mainly consider three types of reasoning traps: *value corruption*, *unit corruption*, and *operation corruption*. The prompt is shown in Prompt 27 and Prompt 28.

## A.6 § 5.1. METACOGNITIVE REASONING SYSTEM

**Implementation Details.** To ensure that the performance gains reported in § 5.1 are attributed to the structural advantage of our metacognitive loop rather than merely increased computational budget (e.g., more token generation or API calls), we established a compute-matched "Vanilla" baseline.

Table 25: Task decomposition: Single turn Prompt.

```
You are a meticulous problem-solving planner.  Your ONLY task is
to create a high-level, step-by-step plan to solve the following
mathematical problem.  The plan should consist of concrete,
actionable steps.  Do NOT actually solve the problem or perform
any calculations.

Problem:
---
{problem_text}
---

Plan:

You are an expert problem solver.  You will be given a problem and
a pre-made plan.  Your task is to follow this plan meticulously to
solve the problem.  Please reason step by step based on the plan.
Finally, provide your final answer within \boxed{}.

Problem:
---
{problem_text}
---

Plan:
---
generated_plan_from_3a
---

Solution:
```

Table 26: Task decomposition: The Prompt for Task Decomposition without CoT.

```
System Prompts:
You are a strategic planner for mathematical problems.  Your goal
is to devise a high-level plan to solve the problem, but you must
NOT execute the detailed calculations yourself.

User Prompts:
Here is the problem:
{problem_text}

Your task is two-fold:
1.  Plan:  Break down the problem into a clear, step-by-step plan.
Describe the strategy and the logical steps required to solve it.
2.  Final Answer:  Based on your intuition of the plan, provide
the final answer immediately.  Do NOT perform step-by-step
calculations or detailed derivations after the plan.

Provide your response in the following format:
Plan:  [Your step-by-step decomposition]
Final Answer:  \boxed{[Your Answer]}
```

Table 27: Prompt of corruption dataset construction (Part 1).

**System Prompts:**
You are a data generator for reasoning robustness evaluation.
Your task is to take an original math/logic problem with its
reasoning process (chain of thought) and answer, and then
intervene in exactly ONE reasoning step with a corruption. The
corruption must be *critical enough to change the final answer*.

**User Prompts:**
Input
Question:
{original_question}

Original Reasoning Process:
{reasoning_steps}

Answer:

{answer}

INSTRUCTION
1. Choose exactly ONE corruption type from the following three
(do not mix):
- Value Corruption: replace a key number with an incorrect but
plausible value
(e.g., change 1 → 1.111111 or g=9.8 → g=1000).
- Unit Corruption: replace the unit of a key step with another
unit
(e.g., meters ↔ centimeters, hours ↔ minutes).
- Operation Corruption: change the mathematical/logical operation
in a key step
(e.g., replace + with −, union with intersection, inequality with
equality).
2. Apply the corruption to ONE critical reasoning step.
- Make sure the corruption influences the correctness of the final
answer.
- After the corruption, truncate the reasoning at that corrupted
step (do not continue to the correct answer).
3. Produce the final output strictly in the following JSON
format:
{
"question": "...",
"corrupted_reasoning": "...", // include the question + steps up
to the corrupted step
"corruption_type": "Value Corruption | Unit Corruption | Operation
Corruption"
}
You are a reasoning evaluator. Your task is to judge whether
a model successfully recognized and adjusted to a corrupted
reasoning trap. You are given:
- The original problem
- The ground truth answer
- The type of corruption applied
- The last corrupted reasoning step (truncated point)
- The model's continued reasoning and final answer

You must decide: Did the model detect and flexibly adjust to the
trap?

Table 28: Prompt of corruption dataset construction (Part 2).

```
System Prompts:
You are a reasoning assistant.  You are given a math/logic
problem, together with a partially completed reasoning process.
Please continue reasoning from what is provided.

User Prompts:
INPUT
Question:  {original_question}

Ground Truth Answer:  {ground_truth_answer}
Ground Truth Corruption Type:  {ground_truth_corruption_type}
Last Corrupted Reasoning Step:  {last_corrupted_step}

Model Generated Reasoning:
model_generated_cot

Model Final Answer:
model_final_answer

TASK Based on the evidence:
- If the model explicitly or implicitly identified the corrupted
reasoning (e.g., points out error, discards it, corrects it)
and produced the correct final answer, output "Yes".
- Otherwise (if it followed the trap blindly, failed to adjust, or
produced the wrong answer), output "No".

Final Output (strict format):
Recognition And Adjustment:  Yes | No
```

Instead of a single-pass direct prompt, our **Vanilla Baseline** is a strong **"Best-of-$K$"** ensemble setup. We prompt the model with the standard instruction: *"Please reason step by step and provide your final answer within \boxed{}."* We set $K = 7$, collecting 7 independent CoT rollouts for each question. This number aligns with the upper bound of the average API calls triggered by our Metacognitive System (which typically terminates within 4–6 steps). The baseline is considered successful if *any* of the $K$ rollouts contain the correct answer (Pass@$K$). Each step were shown in Prompt 31, 32, 33, 34, 35, and 36.

**Ablation Study on Metacognitive Components.** To verify that the performance improvements stem from the *collaboration* of the metacognitive loop components—rather than individual prompting tricks, we conducted an ablation study on the AIME 2025 and GPQA benchmarks using the `DeepSeek-R1` model.

We defined three ablation settings to isolate specific roles: (1) *w/o Planning & Difficulty:* We remove the initial *Planner* role. The system skips the difficulty assessment and task decomposition phase. The first agent immediately generates a reasoning trace, which is then passed directly to the subsequent reasoning/verification agents. (2) *w/o Confidence Monitor:* We disable the explicit *Monitoring* mechanism. The third agent (Verifier) does not generate a structured error report or confidence score. Instead, the reasoning output from the previous step is passed directly to the fourth agent, which attempts a blind correction (standard self-correction) without diagnostic feedback. (3) *w/o Strategy Control:* We remove the *Controller's* ability to adaptively switch strategies. While the Monitor still generates an error report, the system is forced to perform a direct re-reasoning attempt based on the error, rather than pivoting to a new high-level strategy (e.g., switching from algebraic derivation to numerical verification).

**Results.** As observed in Tab. 29, the **Full Metacognitive System** consistently outperforms all ablated variants. Notably, the removal of Strategy Control results in the most significant performance drop compared to the full system. This confirms that *strategic flexibility* is the primary driver of robustness in our framework. Removing the Confidence Monitor also leads to a notable decline. Without accurate monitoring, the system loses the precise trigger required to initiate effective self-

correction, degrading the loop into a less efficient trial-and-error process. These findings demonstrate the necessity of our full metacognitive loop design.

Table 29: Ablation Study of Metacognitive Components. We compare the Full System against the Vanilla baseline and three ablated variants. $\Delta$ denotes the improvement over the Vanilla baseline.

| Method | Pass@1 (AIME) | $\Delta$ | Pass@1 (GPQA) | $\Delta$ |
|---|---|---|---|---|
| Baseline (Vanilla Best-of-7) | 87.5% | - | 81.0% | - |
| (a) w/o Planning & Difficulty | 89.3% | +1.8% | 90.2% | +9.2% |
| (b) w/o Confidence Monitor | 88.9% | +1.4% | 87.5% | +6.5% |
| (c) w/o Strategy Control | 88.1% | +0.6% | 85.3% | +4.3% |
| **Full Metacognitive System** | **90.0%** | **+2.5%** | **93.4%** | **+12.4%** |

In this section, we discuss the future potential of our Metacognitive Reasoning System and present **Me**tacognitive **R**easoning **S**ystem **L**earning (**MeRSL**), a RL method designed to further enhance emergent metacognition system collaboration.

**Enhancing Metacognition Loop for Emergent Metacognition System.** The Prompt-Driven Metacognitive Reasoning System (§ 5.1) demonstrates that explicit role-playing effectively elicits latent reasoning capabilities. However, this "emergent" success relies heavily on the inherent instruction-following and in-context learning abilities of large-scale foundation models (e.g., `Gemini-2.5-Pro`). Smaller models, despite possessing the foundational metacognitive signals identified in § 5.2, often struggle to maintain such complex functional loops via prompting alone. They lack the stability to coordinate distinct roles purely through context.

This limitation motivates us to ask: *Can we optimize the collaborative dynamics between metacognitive and cognitive roles via training?* Building on recent findings that RL can enhance subcomponents of metacognition system (Yang & Thomason, 2025; Wan et al., 2025), MeRSL aims to bridge this gap. By formalizing the interaction between roles as a trainable cooperative game, we can improve our metacognition system.

MeRSL structures the reasoning process as a hierarchical interaction between two distinct policy levels. We decompose the optimization into two sequential stages corresponding to the cognitive loop in Fig. 7: (1) **Stage 1**: Optimizing the collaboration between a Meta-Agent (assessing difficulty and generating plans) and a Reasoning-Agent (executing the solution). (2) **Stage 2**: Optimizing the interaction between a Monitor-Agent (evaluating confidence) and a Strategy-Agent (adjusting the path).

**Stage 1.** We decouple the generation process into a high-level meta-policy $\pi_h$ and a low-level reasoning-policy $\pi_l$. Here, we leverage SFT as a cold start to introduce the high-level agent to various difficulty formats it can utilize.

Formally, The high-level policy ($\pi_h$) first conditions on the input $\mathbf{x}$ to generate a metacognitive directive $\mathbf{m}$ (e.g., `<difficulty>...<plan>...`). Subsequently, the low-level policy $\pi_l$ conditions on both the input and this directive to produce the reasoning trajectory $\mathbf{y}$. The generation process is formulated as:

$$\mathbf{y} \sim \pi_l(\mathbf{y} \mid \mathbf{x}, \mathbf{m})\, \pi_h(\mathbf{m} \mid \mathbf{x}). \tag{8}$$

*Optimization.* During training, suppose $\theta_h$ and $\theta_l$ denote the parameters for the high-level and low-level agents, respectively. The joint system policy $\pi_{(\theta_h, \theta_l)}$ is formlated as:

$$\mathbf{y} \sim \pi_{(\theta_h, \theta_l)}(\mathbf{y} \mid \mathbf{x}) := \pi_{\theta_l}(\mathbf{y} \mid \mathbf{x}, \mathbf{m}) \cdot \pi_{\theta_h}(\mathbf{m} \mid \mathbf{x}), \tag{9}$$

The objective is to maximize the expected reward $R(\mathbf{y}, \mathbf{y}^*)$, which defined as:

$$\mathcal{J}(\theta_h, \theta_l) = \mathbb{E}_{\mathbf{x}, \mathbf{y}^*} \mathbb{E}_{\mathbf{y} \sim \pi_{(\theta_h, \theta_l)}} R(\mathbf{y}, \mathbf{y}*). \tag{10}$$

We adopt an iterative optimization strategy where each agent maximizes their respective rewards independently. The optimization is decoupled as follows:

$$\theta_h^* = \arg\max_{\theta_h} \mathbb{E}_{(\mathbf{x},\mathbf{y}^*)\sim\mathcal{D},\mathbf{m}\sim\pi_{\theta_h},\mathbf{y}\sim\pi_{\theta_l^*}} \left[ R_h(\mathbf{m},\mathbf{y},\mathbf{y}^*) \right], \tag{11}$$

$$\theta_l^*(\theta_h) = \arg\max_{\theta_l} \mathbb{E}_{(\mathbf{x},\mathbf{y}^*)\sim\mathcal{D},\mathbf{m}\sim\pi_{\theta_h},\mathbf{y}\sim\pi_{\theta_l}} \left[ R_l(\mathbf{m},\mathbf{y},\mathbf{y}^*) \right], \tag{12}$$

where $R_h$ and $R_l$ are policies' individual reward functions.

*Reward design.* Concretely, the high-level agent is trained to (i) produce metacognitive directives that lead to consistent low-level solutions, (ii) obey the required "`<difficulty>`...`<plan>`..." format without leaking final answers, and (iii) correctly classify the pre-bucketed difficulty. The low-level agent is trained to (i) solve the problem correctly under the given directive, (ii) follow the required reasoning format, and (iii) allocate reasoning budget conditioned on the predicted difficulty.

For high-level agent, the meta-policy $\pi_h$ is evaluated based on the stability of the downstream execution, the accuracy of its self-assessment, and structural compliance.

*Consistency Reward ($R_{cons}$).* Following the intuition that a high-quality plan should lead to unambiguous execution, we reward the meta-agent for reducing the entropy of the low-level agent's output distribution. Given a directive $\mathbf{m}$, we sample $K$ rollouts $\{\mathbf{y}^{(k)}\}_{k=1}^K$ from $\pi_l(\cdot \mid \mathbf{x},\mathbf{m})$ and extract their final answers $a^{(k)}$. The consistency reward is defined as the empirical majority vote ratio:

$$R_{\text{cons}}(\mathbf{m}) = \max_a \frac{1}{K} \sum_{k=1}^K \mathbb{I}\left[a^{(k)} = a\right]. \tag{13}$$

This objective encourages $\pi_h$ to generate directives that guide $\pi_l$ toward a stable solution mode, mitigating reasoning variance.

*Difficulty Calibration Reward ($R_{diff}$).* To ground the agent's metacognition in objective standards, we introduce a calibration term. Let $d^* \in \{\texttt{easy}, \texttt{medium}, \texttt{hard}\}$ denote the ground-truth difficulty bucket, and $\hat{d} = g(\mathbf{m})$ be the predicted difficulty parsed from the directive. We apply a binary reward for correct classification:

$$R_{\text{diff}}(\mathbf{m}) = \mathbb{I}\left[\hat{d} = d^*\right]. \tag{14}$$

Weighted by $\lambda = 0.2$, this term explicitly aligns the model's internal assessment with the dataset's complexity distribution without dominating the planning objective.

*Format Regularization ($R_{fmt}^h$).* We enforce structural adherence via a format reward $R_{\text{fmt}}^h$. The agent receives a positive signal for correctly generating the `<difficulty>` and `<plan>` tags. Crucially, to enforce the abstraction boundary between planning and solving, the agent incurs a severe penalty if it generates solution-specific artifacts (e.g., \boxed{...}) within the planning phase.

The total high-level reward is aggregated as:

$$R_h(\mathbf{m},\mathbf{y},\mathbf{y}^*) = R_{\text{cons}}(\mathbf{m}) + \lambda R_{\text{diff}}(\mathbf{m}) + R_{\text{fmt}}^h(\mathbf{m}). \tag{15}$$

For low-level agent, the reasoning policy $\pi_l$ focuses on solving the problem correctly, but with a novel constraint: it must allocate computational resources commensurate with the assessed difficulty.

*Correctness Reward ($R_{cor}$).* The primary learning signal remains the binary correctness of the final answer against the ground truth $f(\mathbf{y}^*)$:

$$R_{\text{cor}}(\mathbf{y},\mathbf{y}^*) = \mathbb{I}\left[f(\mathbf{y}) = f(\mathbf{y}^*)\right]. \tag{16}$$

*Difficulty-Adaptive Length Penalty ($R_{len}$).* To realize metacognitive control over the reasoning budget, we introduce a difficulty-conditioned length penalty. Let $T(\mathbf{y})$ be the token length of the reasoning trace and $T_{\text{ref}}$ be a reference length (e.g., the average CoT length in the training set). The

penalty is modulated by the predicted difficulty $\hat{d}$:

$$R_{\text{len}}(\mathbf{m}, \mathbf{y}) = -\alpha(\hat{d}) \cdot \frac{T(\mathbf{y})}{T_{\text{ref}}}, \quad \text{where} \quad \alpha(\hat{d}) = \begin{cases} 0.2, & \text{if } \hat{d} = \texttt{easy}, \\ 0.1, & \text{if } \hat{d} = \texttt{medium}, \\ 0, & \text{if } \hat{d} = \texttt{hard}. \end{cases} \quad (17)$$

This piecewise function $\alpha(\hat{d})$ serves as a soft constraint: it discourages verbosity for simple problems (efficiency) while effectively uncapping the reasoning budget for hard problems (exploration), aligning compute allocation with the task's intrinsic complexity.

*Format Regularization ($R_{fmt}^l$).* A lightweight term $R_{\text{fmt}}^l$ rewards the presence of valid reasoning delimiters (e.g., `<think>...</think>`) and penalizes malformed traces, ensuring the stability of the parsing and evaluation pipeline.

The total low-level reward is summarized as:

$$R_l(\mathbf{m}, \mathbf{y}, \mathbf{y}^*) = R_{\text{cor}}(\mathbf{y}, \mathbf{y}^*) + R_{\text{len}}(\mathbf{m}, \mathbf{y}) + R_{\text{fmt}}^l(\mathbf{y}). \quad (18)$$

**Stage 2.** After Stage 1 establishes the meta–reasoning hierarchy, we further optimize the collaboration between the *Monitor-Agent* (confidence/error evaluation) and the *Strategy-Agent* (trajectory adjustment). Concretely, we instantiate three role-conditioned policies from the same underlying LRM: a reasoning policy $\pi_{\theta_l}$, a monitor policy $\pi_{\theta_m}$, and a strategy policy $\pi_{\theta_s}$. Given the input $\mathbf{x}$ and the metacognitive directive $\mathbf{m}$ produced in Stage 1, the single-round correction process is:

$$\mathbf{y} \sim \pi_{\theta_l}(\mathbf{y} \mid \mathbf{x}, \mathbf{m}), \quad \mathbf{e} \sim \pi_{\theta_m}(\mathbf{e} \mid \mathbf{x}, \mathbf{m}, \mathbf{y}), \quad \mathbf{y}' \sim \pi_{\theta_s}(\mathbf{y}' \mid \mathbf{x}, \mathbf{m}, \mathbf{y}, \mathbf{e}), \quad (19)$$

where $\mathbf{y}$ is the initial reasoning trajectory, $\mathbf{e}$ is the monitor report, and $\mathbf{y}'$ is the corrected trajectory. We perform only *one* Monitor→Strategy correction round in training.

*Grouped rollouts.* For each training instance $(\mathbf{x}, \mathbf{m}, \mathbf{y}^*)$, we first sample $G$ rollout:

$$\{\mathbf{y}^{(i)}\}_{i=1}^G, \quad \mathbf{y}^{(i)} \sim \pi_{\theta_l}(\cdot \mid \mathbf{x}, \mathbf{m}). \quad (20)$$

Each $\mathbf{y}^{(i)}$ is evaluated by a correctness reward defined in Eq. 16

*Priority-Guided Seed Sampling.* To train the Monitor-Agent on informative cases, we select $K$ seeds from the reasoning group using *Priority-Guided Seed Sampling (PGSS)*. PGSS prioritizes incorrect trajectories: (1) if at least $K$ trajectories have $R_l^{(i)} = 0$, we uniformly sample $K$ seeds from them; (2) otherwise, we take all incorrect trajectories and uniformly sample the remaining seeds from the correct ones. Denote the selected reasoning seeds as $\{\tilde{\mathbf{y}}^{(k)}\}_{k=1}^K$.

For each seed $\tilde{\mathbf{y}}^{(k)}$, the Monitor-Agent samples a report group of size $G$:

$$\{\mathbf{e}^{(k,j)}\}_{j=1}^G, \quad \mathbf{e}^{(k,j)} \sim \pi_{\theta_m}(\cdot \mid \mathbf{x}, \mathbf{m}, \tilde{\mathbf{y}}^{(k)}). \quad (21)$$

Each report contains a judgment $\hat{z}^{(k,j)} \in \{\texttt{OK}, \texttt{ERROR}\}$ indicating whether the seed trajectory is believed to be correct.

*Monitor reward.* Let $z^{(k)} = \mathbb{I}[f(\tilde{\mathbf{y}}^{(k)}) = f(\mathbf{y}^*)]$ be the true correctness of the seed. We define a local monitoring reward,

$$R_{\text{loc}}^{(k,j)} = \begin{cases} +1, & \hat{z}^{(k,j)} = \texttt{OK} \text{ and } z^{(k)} = 1, \\ +1, & \hat{z}^{(k,j)} = \texttt{ERROR} \text{ and } z^{(k)} = 0, \\ -1, & \text{otherwise}, \end{cases} \quad (22)$$

encouraging accurate confidence/error judgments.

We further introduce a *correction-gain reward* to align monitoring with downstream usefulness. From the reports that predict $\texttt{ERROR}$, we select up to $K$ report seeds using *Reliability-Guided Report Sampling (RGRS)*: we first prioritize reports with $R_{\text{loc}}^{(k,j)} = +1$, and fill any remaining slots with other $\texttt{ERROR}$ reports. If no $\texttt{ERROR}$ report is produced, the correction branch terminates early. Denote a selected report seed as $\tilde{\mathbf{e}}^{(k,m)}$.

Table 30: Stage-wise results with a single correction round at inference.

| Dataset | Prompt-based Loop | Stage 1 | Stage 2 | Stage 1 + Stage 2 |
|---------|-------------------|---------|---------|-------------------|
| MATH500 | 75.3 | 76.2 | 77.2 | **79.4** |
| GSM8K | 92.2 | 91.8 | 92.6 | 92.6 |

For each selected $\tilde{\mathbf{e}}^{(k,m)}$, the Strategy-Agent samples $G$ corrected trajectories:

$$\{\mathbf{y}'^{(k,m,\ell)}\}_{\ell=1}^{G}, \quad \mathbf{y}'^{(k,m,\ell)} \sim \pi_{\theta_s}(\cdot \mid \mathbf{x}, \mathbf{m}, \tilde{\mathbf{y}}^{(k)}, \tilde{\mathbf{e}}^{(k,m)}), \tag{23}$$

with correctness reward

$$R_s^{(k,m,\ell)} = \mathbb{I}\Big[f(\mathbf{y}'^{(k,m,\ell)}) = f(\mathbf{y}^*)\Big]. \tag{24}$$

Let the correction success ratio be

$$p^{(k,m)} = \frac{1}{G} \sum_{\ell=1}^{G} R_s^{(k,m,\ell)}. \tag{25}$$

The gain reward for the selected monitor report is

$$R_{\text{gain}}^{(k,m)} = \begin{cases} p^{(k,m)}, & z^{(k)} = 0 \ \ (\text{incorrect seed improved}), \\ -(1 - p^{(k,m)}), & z^{(k)} = 1 \ \ (\text{correct seed degraded}). \end{cases} \tag{26}$$

Unselected reports receive $R_{\text{gain}} = 0$. The total monitor reward is

$$R_m^{(k,j)} = R_{\text{loc}}^{(k,j)} + R_{\text{gain}}^{(k,j)}. \tag{27}$$

*Optimization.* We update the three role policies using GRPO. For each group of size $G$ generated under the same conditioning context, we compute normalized intra-group advantages

$$\hat{A}_i = \frac{R_i - \text{mean}(R_{1:G})}{\text{std}(R_{1:G}) + \epsilon}, \tag{28}$$

and apply a clipped policy-gradient update with KL regularization to the corresponding role. Reasoning groups use rewards $\{R_l^{(i)}\}$, monitor groups use $\{R_m^{(k,j)}\}$, and strategy groups use $\{R_s^{(k,m,\ell)}\}$. This joint training encourages the Monitor-Agent to produce diagnostically useful feedback and the Strategy-Agent to reliably correct low-confidence trajectories, thereby strengthening the emergent metacognitive loop under limited model capacity.

**Details.** We conduct training on `Qwen2.5-7B-Instruct` using the `MATH` dataset. We optimize the model with Adam Optimizer using a constant learning rate of $1e-6$. During rollout, the prompt batch size is set to 128, and we sample $G = 8$ responses for each prompt. The sampling temperature is 1 with `top-p = 1.0` and `top-k = -1`. We cap the maximum response length at 2048 tokens. For stage 1, we set $\lambda = 0.2$ and $T_{ref} = 1024$. For stage 2, we set seeds $K = 2$.

**Results.** Under the constrained inference setting where only a single correction round is allowed, we evaluate the effectiveness of Stage 1, Stage 2, and their combination. All experiments are conducted with `Qwen2.5-7B-Instruct` on two standard mathematical reasoning benchmarks: `MATH500` and `GSM8K`. We use the instruct model to ensure better instruction-following behavior.

On `GSM8K`, Stage 1 yields a mild degradation. We conjecture this is because GSM8K problems are relatively simple and short-horizon; explicit planning and difficulty self-assessment provide limited additional benefit, while introducing a small overhead that can occasionally distract the low-level execution. In contrast, Stage 2 consistently improves GSM8K, suggesting that even for easier problems, a monitor–strategy correction step can fix local arithmetic slips. Combining Stage 1 and Stage 2 maintains the gain from Stage 2 without further improvement, implying that Stage 2 dominates the achievable headroom under the one-round inference constraint. Overall, these results demonstrate that our trainable metacognitive collaboration yields robust gains on difficult mathematical reasoning, while remaining effective under strict inference budgets.

Table 31: Metacognitive Reasoning System: Prompt of Step 1.

**System Prompt:**
You are an elite mathematical strategist and analyst. Your
primary function is to perform a deep Metacognitive Analysis of
complex mathematical problems. You are to deconstruct the problem
into its core components, identify underlying principles, and then
formulate a high-level, executable strategic plan.
Your task is to produce a Metacognitive Analysis of the following
problem. You must NOT provide a final solution or perform
detailed calculations.
### Core Principles
* **Analytical Depth:** Your analysis must go beyond a
surface-level reading. Identify the mathematical field, key
concepts, constraints, and the explicit goal.
* **Strategic Foresight:** Your plan should be a viable path to
a solution. This includes anticipating potential difficulties,
identifying necessary lemmas, and choosing the most promising
approach.
* **Clarity and Brevity:** The analysis and plan must be clear,
concise, and easily understood by another mathematical expert who
will execute it.
**User Prompt:**
### Your Task
**Problem:**
==========
{problem_statement}
==========
**Metacognitive Analysis:**
**1. Problem Deconstruction:**
* **Mathematical Domain:** Identify the primary field(s) of
mathematics involved (e.g., Number Theory, Combinatorics,
Euclidean Geometry).
* **Given Conditions & Constraints:** List all the premises,
conditions, and constraints provided in the problem statement in
a structured format.
* **Objective:** State the precise question to be answered or the
proposition to be proven.
**2. Strategic Solution Plan (Method Sketch):**
Present a high-level, conceptual outline of your proposed solution
path. This sketch should enable an expert to grasp the entire
logical flow of the argument without needing the full details. It
must include:
* **Overall Strategy Narrative:** A brief description of the core
idea behind your approach (e.g., "We will use proof by induction,"
"The strategy is to establish a coordinate system and use analytic
geometry," "We will prove the contrapositive by assuming...").
* **Key Lemmas and Intermediate Results:** State the full and
precise mathematical formulations of any key lemmas or theorems
you plan to prove or apply. These are the major milestones of the
proof.
* **Logical Skeleton:** If applicable, describe the key
constructions, case splits, or transformations that form the
backbone of your argument.
* **Potential Challenges & Pitfalls:** Briefly note any steps
that might be particularly tricky, prone to error, or require a
non-obvious insight.
### Negative Constraints
* **DO NOT** write the full, step-by-step solution.
* **DO NOT** perform detailed algebraic manipulations or numerical
calculations.
* Your output should be strictly limited to the analysis and
strategic plan as outlined above.

Table 32: Metacognitive Reasoning System: Prompt of Step 2.

**System Prompt:**
You are an exceptionally rigorous mathematical solver. Your
sole purpose is to take a pre-defined strategic plan and execute
it with absolute precision and logical soundness. You must not
deviate from, question, or reinterpret the provided plan.
Your task is to produce a complete and formally justified solution
to the following mathematical problem, strictly following the
`Solution Plan`.
### Core Principles
* **Rigor is Paramount:** Your primary goal is to produce a
complete and rigorously justified solution. Every step in your
solution must be logically sound and clearly explained. A correct
final answer derived from flawed or incomplete reasoning is
considered a failure.
* **Unyielding Adherence to Plan:** You MUST strictly follow
the logical flow, lemmas, and constructions laid out in the
`Solution Plan`. Do not introduce new methods, skip steps, or
alter the proposed strategy in any way. Your role is execution,
not creation.
* **Honesty About Completeness:** If you cannot find a complete
solution following the plan, you must **not** guess or create a
solution that appears correct but contains hidden flaws. Instead,
you should present only the significant partial results that you
can rigorously prove by following the plan.

**User Prompt:**
### Your Task
**Problem:**
==========
{problem_statement}
==========
**Solution Plan:**
==========
{step1_output}
==========
**Detailed Solution:**
Present the full, step-by-step mathematical proof, meticulously
following the guidance of the `Solution Plan`. Each step must be
logically justified and clearly explained. The level of detail
should be sufficient for an expert to verify the correctness of
your reasoning without needing to fill in any gaps. This section
must contain ONLY the complete, rigorous proof, free of any
internal commentary, alternative approaches, or failed attempts.
* **Use TeX for All Mathematics:** All mathematical variables,
expressions, and relations must be enclosed in TeX delimiters
(e.g., `Let $n$ be an integer.`).
Your step-by-step reasoning, strictly following the plan, begins
here...
### Final Answer
After completing the detailed solution, state the final answer
within \boxed{}.

1674
1675
1676
1677
1678

Table 33: Metacognitive Reasoning System: Prompt of Step 3 (Part 1).

1679
1680
1681
1682
1683
1684
1685
1686
1687
1688
1689
1690
1691
1692
1693
1694
1695
1696
1697
1698
1699
1700
1701
1702
1703
1704
1705
1706
1707
1708
1709
1710
1711
1712
1713
1714
1715
1716
1717
1718
1719
1720
1721
1722
1723
1724
1725
1726
1727

**System Prompt:**
You are an expert mathematician and a meticulous grader for
AIME-level computational problems. Your primary task is to
rigorously verify the provided solution's **computational
reasoning and numeric correctness**. A solution is to be judged
correct **only if every step that affects the numeric outcome
is correct and sufficiently justified.** A solution that reaches
a correct final integer answer via arithmetic slips, incorrect
algebraic manipulations, unverified casework, counting mistakes,
or hidden assumptions must be flagged as incorrect or incomplete.
### Instructions ###
**1. Core Instructions**
* Your sole task is to identify and report all issues in the
provided solution. You must act strictly as a **verifier**, NOT
a solver.
* You must **NOT attempt to correct, fix, or complete** any errors
or missing arguments.
* Perform a **step-by-step** check of the entire solution and
produce a **Detailed Verification Log**. For each step:
* If the step is correct, state briefly that it is correct.
* If the step contains an issue, explain the error and classify it
(see section 2).
**2. How to Handle Issues in the Solution**
All issues must be classified into one of the following
categories:
* **a. Critical Error:**
* Definition: Any error that changes or potentially invalidates
the numeric result. Examples include arithmetic mistakes,
wrong algebraic transformations, misapplied formulas, incorrect
combinatorial counts, invalid casework, or unjustified
approximations that affect the integer outcome.
* **Procedure:**
* Point out the exact error and explain why it invalidates the
reasoning.
* Do **not** check further steps that rely on this error.
* You may still check other independent parts of the solution.
* **b. Justification Gap:*** Definition: Steps where the stated
conclusion might be correct, but the reasoning is incomplete or
not justified at AIME level.
* **Procedure:**
* Point out the missing justification.
* Explicitly state that you will assume the step's conclusion
holds for the sake of checking subsequent steps.
**3. Output Format**
Your response MUST be structured into two main sections: a
**Summary** followed by the **Detailed Verification Log**.
* **a. Summary*** **Final Verdict:** One clear sentence declaring
overall validity (e.g., "The solution is correct," "The solution
contains a Critical Error and is therefore invalid," or "The
solution contains several Justification Gaps.").
* **List of Findings:** A bulleted list of every issue found. For
each finding include:
* **Location:** A direct quote of the key phrase or equation.
* **Issue:** Short description and classification (**Critical
Error** or **Justification Gap**).

Table 34: Metacognitive Reasoning System: Prompt of Step 3 (Part 2).

```
 * **b.  Detailed Verification Log*** Provide a step-by-step
verification.
* Quote the relevant part of the solution before your check.
* State clearly:  **Correct**, **Critical Error**, or
**Justification Gap**.
* Do **not** supply corrections or alternative methods | only
report the issues.
**Important:**
- Do not propose fixes or alternative solutions.
- Do not attempt to supply missing reasoning.
- Only check and report correctness of what is written.

User Prompt:
### Your Task
**Original Problem:**
==========
{problem_statement}
==========
**Current Solution:**
==========
{last_solution}
==========

### Monitoring Task Reminder ###
Your task is to act as an math grader.  Now, generate the
**summary** and the **step-by-step verification log** for the
solution above.  In your log, justify each correct step and
explain in detail any errors or justification gaps you find, as
specified in the instructions above."
```

## A.7 § 5.2. METACOGNITIVE REASONING MODELS: MORE DETAILS

**Implementation Details.** For the online model APIs, we utilized their respective official endpoints with default temperature settings. The maximum generation length was also kept at its default value, and no other parameters were modified.

For the training of open-source models, all experiments were conducted on a server equipped with 8×NVIDIA H800 GPU (80GB). The maximum generation length was set to 8192 for the GSM8K and MATH datasets, and 16384 for all other datasets. The temperature was set to 1.0, and the random seed was fixed to 42 for reproducibility.

For SFT training, we adpot full-parameter fine-tuning with learning rate $r = 1 \times 10^{-5}$ and $batchsize = 32$. To construct training data, we first define the difficulty level for each sample from the train set of GSM8K and MATH: Easy (GSM8K and MATH lv.1), Medium (MATH lv.1-4), and Hard (MATH lv.3-5). Second, to obtain high-level task decomposition, we utilize Gemini-2.5-Pro with the prompt in Appendix A.4. The final pattern of training samples in cold-start is in the form of: ``<difficulty> level </difficulty> <plan> decomposition </plan> <think> CoT </think> answer''.

For RL training, we write our code based on the open-source Verl framework. Training settings are listed in Tab. 40. During inference, we used a standard prompt without any task-specific engineering during inference for all models (Base, GRPO, and Ours):*"Please reason step by step and provide your final answer within* $\backslash boxed\{\}$*."*

**Discussion on Signal Selection in RL** In our MRM RL training, we conducted an ablation study on signal selection. We compared our default *Token Confidence* against (1) *Sentence Confidence*, the average confidence of the current sentence. *Sentence Entropy*, The mean entropy of the sentence token. *Random Forking*, A control baseline where forking occurs at random positions until the target group size is reached, decoupling exploration from uncertainty.

Table 35: Metacognitive Reasoning System: Prompt of Step 4 (Part 1).

**System Prompt:**
You are an expert mathematician and a careful corrector for
AIME-level computational problems.
You will be given three inputs:
1) The Original Problem,
2) The current Solution,
3) A Verification Log (from a previous check), which labels each
step as Correct / Justification Gap / Critical Error, and provides
short notes.
### Your Task ###
Using the Verification Log, **step by step correct the Original
Solution**.
- If a step is labeled **Correct**, keep it unchanged (you may
lightly reformat for clarity).
- If a step is labeled **Justification Gap**, supply the missing
justification or intermediate calculations, enough for AIME-level
rigor.
- If a step is labeled **Critical Error**, replace it with
a correct mathematical step (with explicit computations or
reasoning) and update all dependent later steps accordingly.
- Do **not** introduce new solution paths, alternative methods, or
multiple approaches.  Only repair the given solution chain.
### Output Format ###
1.  **Correction Summary**
- A single sentence declaring whether the solution has been fully
corrected and what the final answer is.
- Example:  \The solution has been fully corrected.  Final Answer
= 70."
- Or, if not possible:  \The solution cannot be fully corrected
due to missing information in step X."
2.  **Correction Log**
For each relevant step (especially those flagged in the
Verification Log), provide an entry with:
- **Quoted Step:** The original line/equation (quoted or in a code
block).
- **Verification Label:** Correct / Justification Gap / Critical
Error.
- **Correction / Action:**
* If Correct ➝ \Unchanged | correct."
* If Justification Gap ➝ Provide the missing computation/derivation
briefly, ending with \Filled gap."
* If Critical Error ➝ Provide the corrected computation/derivation,
briefly note why the original was wrong, and end with \Corrected."
- If a step's correction affects later steps, explicitly note
\Affects subsequent steps:  Yes/No."
3.  **Full Corrected Solution**
- Present the entire solution in a clean, continuous write-up,
combining unchanged and corrected steps.
- Show all necessary algebra, arithmetic, or combinatorial
reasoning clearly.
- After completing the detailed solution, state the final answer
within \boxed{}.

Table 36: Metacognitive Reasoning System: Prompt of Step 4 (Part 2).

```
 User Prompt:
### Your Task
***Original Problem:**
==========
{problem_statement}
==========
**Current Solution:**
==========
{last_solution}
==========
**Verification Log:**
==========
{monitor_output}
==========
```

Table 37: Ablation study on RL performance with different monitoring signals. **Random** denotes random forking strategies.

| Metric Type | MATH500 (Pass@1) | GSM8K (Pass@1) |
|---|---|---|
| Qwen2.5-Math-7B (Base) | 64.0% | 70.3% |
| *Baseline (GRPO)* | 71.6% | 75.9% |
| Random Forking | 66.3% | 71.4% |
| Sentence Confidence | 72.4% | 79.1% |
| Sentence Entropy | 73.0% | 78.8% |
| **Token Confidence (Ours)** | **80.2%** | **85.5%** |

**Results.** As shown in Tab. 37, both sentence-level confidence and entropy outperform the standard GRPO baseline, indicating that the uncertainty estimation is robust across different signal definitions. We also find the random setting leads to performance degradation compared to GRPO. This negative result is crucial: it confirms that simply increasing exploration diversity is insufficient and can be detrimental. Overall, these results validate that the primary driver of improvement is our metacognitive control mechanism.

Table 38: Performance (%) on Out-of-Distribution (OOD) Benchmarks. **Ours (SFT+RL)** demonstrates superior generalization capabilities compared to baselines.

| Method | ARC-c | GPQA-Diamond | MMLU-Pro | LiveCodeBench |
|---|---|---|---|---|
| Base Model (Qwen2.5-Math-7B) | 18.2 | 33.8 | 37.4 | 28.9 |
| GRPO (Standard RL) | 29.8 | 39.2 | 42.1 | 35.5 |
| **Ours (SFT+RL)** | **33.2** | **46.5** | **47.6** | **40.4** |

**Out-of-Distribution (OOD) Generalization.** To validate generalization, we evaluated our fine-tuned model (*SFT+RL*) against the Qwen2.5-Math-7B and the standard RL baseline (*GRPO*) on four diverse OOD benchmarks: ARC-C (Clark et al., 2018): A challenging open-domain reasoning benchmark. GPQA-DIAMOND: A graduate-level science benchmark covering biology, physics, and chemistry. MMLU-PRO (Wang et al., 2024): A comprehensive benchmark focusing on complex reasoning across diverse academic subjects. LIVECODEBENCH (Jain et al., 2024): A holistic evaluation for code generation, representing a significant domain shift from mathematics. To avoid contamination, we shuffled the multiple-choice options for all QA tasks.

**Results.** The performance comparison is presented in Tab. 38. Our method consistently outperforms both the Base Model and the GRPO baseline across all benchmarks. Specifically, our method achieves substantial gains on scientific reasoning and general academic reasoning. This confirms that the internalized metacognitive skills like *planning* are not merely overfitting to math problems but can transfer effectively to general reasoning tasks. On LIVECODEBENCH, our method shows

a modest but meaningful improvement. We attribute the relative difficulty of this task to the base model architecture (`Qwen2.5-Math`), which is specialized for mathematics rather than coding.

**Integration with Self-Improvement Paradigms.** A critical extension regarding our framework is its relationship with self-training methods like STaR (Zelikman et al., 2024). we conducted a controlled experiment combining our Metacognitive Schema with the STaR iterative loop. We utilized `Qwen2.5-Math-7B` as the base model and evaluated performance on the MATH-500 benchmark across multiple iterations. The comparison settings were *Standard STaR:* Iteratively fine-tuning on standard rationales: $Q \rightarrow \texttt{CoT} \rightarrow A$, and our *Meta-STaR:* Iteratively fine-tuning on structured metacognitive traces: $Q \rightarrow \texttt{<difficulty>} \rightarrow \texttt{<plan>} \rightarrow \texttt{<think>} \rightarrow A$.

Table 39: Performance comparison on MATH-500 across bootstrapping iterations. **Meta-STaR** demonstrates a faster rate of improvement compared to standard STaR.

| Method | Iteration 0 (Base) | Iteration 2 | Iteration 4 |
|---|---|---|---|
| Qwen2.5-Math-7B | 64.0% | - | - |
| Standard STaR | 64.0% | 65.7% | 67.0% |
| **Meta-STaR (Ours)** | 64.0% | **68.6%** | **71.2%** |

**Results.** As detailed in Tab. 39, while standard STaR yields consistent improvements, applying the STaR algorithm to our metacognitive schema results in significantly higher gains. This substantial gap suggests that incorporating metacognitive signals makes the bootstrapping process more efficient. By enforcing explicit planning and self-assessment, the model generates higher-quality rationales during the exploration phase, thereby creating superior training data for subsequent iterations. This confirms that our structured metacognitive paradigm can effectively serve as a foundational architecture for advanced self-improvement algorithms.

## A.8 Statement on the Use of Large Language Models (LLMs)

**LLMs as the Subject of Research.** One of the core components of our research involves the investigation and evaluation of the reasoning capabilities of current open-source and closed-source Large Language Models (LLMs). As such, a number of LLMs are explicitly named and analyzed within this paper (as detailed in § 4). In this capacity, they serve as the objects of our study.

**LLMs as an Assistive Tool.** In the preparation of this manuscript, the use of LLMs is limited to polishing the text for grammatical correctness, spelling, and clarity of expression. The LLMs were not used to generate any core research ideas, experimental designs, data analysis, or substantive portions of the manuscript.

We assume full responsibility for all content presented in this paper, including any text that has been revised with the assistance of an LLM. We have meticulously reviewed and edited all content to ensure its scientific accuracy and originality, preventing any form of plagiarism or academic misconduct.

Table 40: Training settings for RL.

| Parameter | Value |
| --- | --- |
| n_gpu | 8 |
| rollout.n | 16 |
| total_steps | 1000 |
| batch_size | 8 |
| critic_warmup | 0 |
| max_prompt_length | 512 |
| max_response_length | 16384 |
| filter_overlong_prompts | True |
| learning_rate | 1e-6 |
| use_kl_loss | True |
| kl_loss_coef | 0.001 |
| kl_loss_type | low_var_kl |

