# OpenReview forum: "Towards Understanding Metacognition in Large Reasoning Models"
_ICLR.cc/2026/Conference — Submitted to ICLR 2026_

### Official Review · Reviewer_DtKq · 2025-10-30

**Soundness:** 2
**Presentation:** 3
**Contribution:** 2
**Rating:** 4
**Confidence:** 3

**Summary:**

This paper investigates metacognition in LRMs. The paper first performs a series of empirical evaluations to measure metacognitive abilities in current LRMs, and then proposes two paradigms for enhancing models’ metacognitive abilities (through test-time prompting as well as parameter-level training).

**Strengths:**

The topic of metacognition is important and timely, and I think the paper makes interesting contributions, both in evaluating metacognition and improving metacognitive abilities.

**Weaknesses:**

I had a few concerns about the design of the self-awareness study in Section 4.1. The authors write that they “operationalize self-awareness as the model’s ability to accurately classify the difficulty of mathematical problems when explicitly prompted to do so” (l. 208-209). They find that the tested LRMS “exhibit a lack of capacity in perceiving task difficulty” (l. 240-241). There are several potential issues with this design. First, what do labels like “easy”, “medium”, and “hard” mean for models? Would humans even be able to perform this labeling task? Second, this task may not be a good measure of self-awareness, because there may be confounding factors that can be used to distinguish between the difficulty classes—for example, maybe the hard problems are longer on average, or contain certain key words or phrases that are not present in easy problems. It would be useful to see some control analyses here. Third, and related to the second point, Song et al. (2025) argue that true metacognition requires *privileged self-access*. If model B could predict whether a math problem will be easy/medium/hard for model A as well as model A can predict itself, then model A is not truly showing signs of self-awareness. This is a critical theoretical point that is not discussed anywhere in the paper.

Building off the point above, there are many related works that are not discussed. For example, on the topic of introspection and self-awareness: Binder et al. (2024; “Looking Inward: Language Models Can Learn About Themselves by Introspection”), Song et al. (2025a; “Language Models Fail to Introspect About Their Knowledge of Language”), (2025b; “Privileged Self-Access Matters for Introspection in AI”).

In addition, the importance of intermediate planning has also been extensively studied (e.g., Valmeekam et al. 2023: “On the Planning Abilities of Large Language Models - A Critical Investigation”; Webb et al. 2025: “A brain-inspired agentic architecture to improve planning with LLMs”), so I’m unsure about the novelty of the task decomposition experiment (Section 4.3).

Finally, the paper has no discussion of limitations, and the discussion is very short.

**Questions:**

Could you address the points raised above in the Weaknesses section?

---

> ### Author Response · Authors · 2025-11-23
> **Response to Reviewer DtKq (1/4)**
>
> Thanks for your valuable feedback to help us improve our paper! We have revised our paper based on your feedback. We detail our response below and please kindly let us know if our response addresses your concerns.
>
> ***
>
> >**W1:  Concerns about the definition of difficulty labels and human feasibility.**
>
> We very thank the reviewer for raising this insightful concern regarding the selection and validity of our difficulty labels. We apologize for not elaborating on the specific labeling details in the main text. We provide a detailed clarification below:
>
> **(1) What do labels mean**
> First, our labels (Easy/Medium/Hard) are derived from the numerical ratings that have already been provided in the DeepMath-103K dataset [1]. During the data construction of DeepMath, each question's rating is grounded in the **Art of Problem Solving (AoPS)** difficulty scale [2], which serves as a gold standard in the mathematics competition community. These ratings were generated by an ensemble of GPT-4o (six times) to ensure robust alignment with human expert criteria.
>
> Based on these numerical ratings and the data sources, we categorize the difficulty levels as follows:
>
> - **Easy (Rating < 3.5):** standard high school math competition level problems (e.g., problems similar to the MATH [3]).
> - **Medium (3.5$\leq$ Rating $\leq$ 6.5):** preeminent math competition for K-12 students (e.g., AMC23[4]).
> - **Hard (Rating>6.5):** highly selective invitational or Olympiad-level problems (e.g., AIME2025 [5]).
>
> **(2) Human Feasibility**
> Since the ground truth is derived from the AoPS scale, humans are indeed able to perform this task. For example, It would be easy for human expert to distinguish difference between a level 3 problem *"What is the least positive integer multiple of 30 that can be written with only the digits 0 and 2?"* and a level 5 problem *"Using only 3, 5, and 9 paise coins, what is the largest amount that cannot be paid in exact change?"*.
>
> What's more, to ensure a rigorous evaluation and mitigate ambiguity for **boundary cases**, we implemented two strategies:
>
> - **Relaxed Boundary Evaluation:** For samples with ratings at the decision boundaries (e.g., exactly 3.5), predictions of either adjacent category (e.g., "Easy" or "Medium") are considered correct.
> - **Few-shot Guidance:** In our prompts, we included representative few-shot examples to provide the model with a concrete, intuitive understanding of the "Easy/Medium/Hard" label.
>
> **(3) Are these labels meaningful for models**
> Following the approach in [1], here we conduct an experiment to assess the mathematical performance of the tested models on different level benchmark(subset from DeepMath).
>
> **Table 1: Pass@1 Accuracy of Qwen-2.5-Math-7B across benchmarks of increasing difficulty.**
>
> | Model                | MATH 500 | AMC 23 | Olympiad Bench | AIME 24 | AIME 25 |
> | :------------------- | :------: | :----: | :------------: | :-----: | :-----: |
> | **Qwen-2.5-Math-7B** |   64.0   |  31.9  |      15.8      |  11.2   |   4.4   |
>
> As shown in Table 1, model performance (Pass@1) drops as the benchmark difficulty increases (e.g., from MATH-500 to AMC to Olympiad Bench). This negative correlation suggests that our difficulty labels effectively reflect the performance boundary of the models, making them a meaningful proxy for self-awareness.
>
> We have revised the paper and put these details in **Appendix A.2**.
>
> **References**
>
> [1] *DeepMath-103K: A Large-Scale, Challenging, Decontaminated, and Verifiable Mathematical Dataset for Advancing Reasoning.* arXiv:2504.11456
>
> [2] *https://artofproblemsolving.com/wiki/index.php/AoPS_Wiki:Competition_ratings*
>
> [3] *Measuring Mathematical Problem Solving With the MATH Dataset.* NIPS2021.
>
> [4] *American mathematics competitions (AMC 10/12). Mathematics Competition Series.*
>
> [5] *American invitational mathematics examination (AIME). Mathematics Competition Series.*

---

> ### Author Response · Authors · 2025-11-23
> **Response to Reviewer DtKq (2/4)**
>
> ***
>
> >**W2:  Control Analyses on Confounding Factors (Length, Keywords, and Topics).**
>
> We appreciate the reviewer’s suggestion regarding potential confounding factors!
>
> To verify that our results reflect a real lack of metacognitive self-awareness rather than heavy reliance on spurious correlations, we performed three targeted control analyses.
>
> **(1) Length Analysis**
> The reviewer correctly points out that harder problems might be longer. We calculated the mean and standard deviation of word numbers for problems across difficulty levels 3 to 9.
>
> **Table 2: Token Length Statistics across Difficulty Levels.**
>
> | Level    | Level 3 | Level 4 | Level 5 | Level 6 | Level 7 | Level 8 | Level 9 |
> | :------- | :-----: | :-----: | :-----: | :-----: | :-----: | :-----: | :-----: |
> | **Mean** |  30.0   |  29.6   |  31.0   |  32.6   |  30.3   |  40.1   |  42.87  |
> | **Std**  |  23.3   |  18.6   |  20.2   |  19.9   |  22.1   |  26.0   |  29.4   |
>
> As shown in Table 2, the length distributions for Levels 3–7 significantly overlap. we also observed that a distinct increase in length and variance in Levels 8 and 9.
>
> **Control Experiment:** To rigorously rule out length as a shortcut, we re-evaluated the **Difficulty Assessment Accuracy** on a subset **excluding Level 8 and 9**.
>
> **Table 3: Accuracy Comparison (Original vs. Length-Controlled).**
>
> | Model                | Gemini-2.5-Pro | GPT-OSS-120B | Seed-1.5-VL-Pro | Qwen3-8B | Qwen3-32B |
> | :------------------- | :------------: | :----------: | :-------------: | :------: | :-------: |
> | **Original Acc**     |     58.5%      |    69.0%     |      56.0%      |  36.8%   |   44.9%   |
> |  **Controlled Acc**  |     57.8%      |    67.9%     |      55.2%      |  34.1%   |   41.9%   |
>
> Result: The accuracy drops slightly (approx. -0.9% on average) but remains within a comparable range. This indicates that even when the "length heuristic" (long = hard) is removed, models still exhibit the same fundamental lack of capability in distinguishing difficulty. The failure is intrinsic, not an artifact of length.
>
> **(2) Keywords Analysis**
> Here we consider do hard problems contain specific "trigger words"? We extracted the Top-50 most frequent content words for each level and calculated the **Jaccard Similarity** of these keyword sets between levels.
> J(A,B)=∣A∪B∣∣A∩B∣​
>
>
>
> **Table 4: Jaccard Similarity of Top-K frequent words between Difficulty Levels.**
>
> |              | Easy (L3-4) | Medium (L5-6) | Hard (L7-9) |
> | :----------- | :---------: | :-----------: | :---------: |
> |  **Easy**    |    1.00     |       -       |      -      |
> |  **Medium**  |    0.73     |     1.00      |      -      |
> |  **Hard**    |    0.69     |     0.82      |    1.00     |
>
> If specific trigger words existed, we would expect the keyword sets of different levels to diverge, resulting in **low Jaccard similarity**. Instead, as shown in Tab.4, we observe consistently **moderate-to-high Jaccard similarities** across levels, suggesting substantial overlap in their content words. This indicates that different difficulty levels do not rely on distinct lexical markers; difficulty is not driven by specific keywords but rather by structural or reasoning complexity.
>
> **(3) Topic Analysis**
> To ensure that difficulty assessment isn't confounded by topic (e.g., "Calculus is always Hard"), we analyzed self-awareness accuracy within the top-3 mathematical domains from DeepMath-103K: **Calculus, Algebra, and Precalculus**.
>
> **Table 5: Difficulty Assessment Accuracy Breakdown by Topic.**
>
> | Topic           | Gemini-2.5-Pro | GPT-OSS-120B | Seed-1.5-VL-Pro | Qwen3-8B | Qwen3-32B |
> | :-------------- | :------------: | :----------: | :-------------: | :------: | :-------: |
> | **All Topic**   |     58.5%      |    69.0%     |      56.0%      |  36.8%   |   44.9%   |
> | **Algebra**     |     61.1%      |    68.5%     |      54.7%      |  39.4%   |   47.2%   |
> | **Calculus**    |     56.8%      |    70.4%     |      55.3%      |  37.2%   |   45.9%   |
> | **Precalculus** |     58.3%      |    67.1%     |      58.2%      |  36.0%   |   46.5%   |
>
> As shown in Table 5, we can see the performance is consistent across different topics. Models struggle to assess difficulty within a domain just as much as they do globally. This confirms that our findings reveal a general deficit in metacognition, independent of the specific mathematical field.
>
> **Conclusion:**
> These control analyses demonstrate that neither length, lexical cues, nor topics can explain the models' behavior. The inability to accurately assess difficulty persists even when these factors are controlled, reinforcing our claim that current LRMs lack robust self-awareness. We have revised the paper and put these details in **Appendix A.2**.

---

> ### Author Response · Authors · 2025-11-23
> **Response to Reviewer DtKq (3/4)**
>
> ***
>
> >**W3:  Self-awareness: Is it truly showing signs of self-awareness？**
>
> We thank the reviewer for raising this critical theoretical point! We agree that distinguishing **privileged self-access** from clever surface predicting is important to establishing complete self-awareness.
>
> To address this concern and validate our findings in §4.1, we conducted a new controlled experiment in **Appendix A.2**. There are some main results.
>
> **Experiment:**
> Here we compared **Model A** predicting whether a math problem easy/medium/hard with **Model B** predicting difficulties for Model A, Specifically, we set two groups:
> 1.  **Self-Prediction:** The Model A (subject) assesses the difficulty of a problem for **itself**.
> 2.  **Cross-Model Prediction:** The Model B (observer) is provided with the same problem and few-shot examples and asked to assess the difficulty for Model A. Here, B $\neq$ A.
>
> We choose Qwen2.5-Math-7B,  Qwen3-8B, and Llama-3.1-8B to analyze and judge the math problem difficulties. We evaluated the predicted label consistency and acc on 600 test sampled problems. The details are as follows:
>
> **(1) Consistency on Difficulty Labels:**
> First, we measured the predicted label consistency between Model A and B. If difficulty awareness were truly come from Model A, we would expect low consistency. The results were shown in Table 1.
>
> **Table 6: Difficulty Assessment Consistency**. The off-diagonal section is the cross-model setting, while the bolded diagonal section is the self-reflection setting.
>
> | Model B\Model A     | **Qwen2.5-Math-7B** | **Qwen3-8B** | **Llama-3.1-8B** |
> | :------------------ | :------------------ | :----------- | ---------------- |
> | **Qwen2.5-Math-7B** | **36.38%**          | 41.93%       | 32.57%           |
> | **Qwen3-8B**        | 32.92%              | **46.09%**   | 31.56%           |
> | **Llama-3.1-8B**    | 30.59%              | 24.89%       | **38.74%**       |
> Mean of self-reflection setting: 0.4040
> Mean of cross-model setting: 0.3241
>
> **Results:**
> As shown in Table 6, we find a distinct misalignment between self-prediction and cross-model prediction. The results strongly support the existence of privileged self-access in our tested LRMs.
>
> **(2) Accuracy:**
> We also compared the average accuracy for self-prediction and cross-model prediction. Following [1], we argue that if an LRM A has privileged access to its knowledge, then it should be able to perform better than another model predicting for A.
>
> **Table 7: Accuracy of Difficulty Assessment relative to Subject's Performance**
>
> | Model B      | Model A             | Assessment Accuracy (%) |
> | :----------- | :------------------ | :---------------------- |
> | -            | **Qwen2.5-Math-7B** | **39.2%**               |
> | Llama-3.1-8B | Qwen2.5-Math-7B     | 33.0%                   |
> | Qwen3-8B     | Qwen2.5-Math-7B     | 34.1%                   |
>
> | Model B         | Model A      | Assessment Accuracy (%) |
> | :-------------- | :----------- | :---------------------- |
> | -               | **Qwen3-8B** | **35.7%**               |
> | Llama-3.1-8B    | Qwen3-8B     | 31.1%                   |
> | Qwen2.5-Math-7B | Qwen3-8B     | 29.6%                   |
>
> | Model B         | Model A          | Assessment Accuracy (%) |
> | :-------------- | :--------------- | :---------------------- |
> | -               | **Llama-3.1-8B** | **47.0%**               |
> | Qwen2.5-Math-7B | Llama-3.1-8B     | 39.7%                   |
> | Qwen3-8B        | Llama-3.1-8B     | 42.7%                   |
>
> | Setting                | Avg Accuracy (%) |
> | :--------------------- | ---------------- |
> | Self-Prediction        | 40.6%            |
> | Cross-Model Prediction | 35.0%            |
>
> **Results:**
> As shown in Table 7, we can notice that self-prediction accuracy is much higher than across-model prediction, while it still exhibits relatively low capacity for self-awareness. These results suggest that LRMs are likely to use privileged self-access to assess task difficulties.
>
> **Conclusion:**
> The results, where Model B fail to predict the difficulty as accurately as the subject Model A itself, demonstrates that **our difficulty assessment setting captures true self-awareness**. We have added this discussion to the revised paper (**Appendix A.2**) to strengthen our theoretical claims.

---

> ### Author Response · Authors · 2025-11-23
> **Response to Reviewer DtKq (4/4)**
>
> ***
>
> >**W4:  Discussion of Related Work**
>
> We sincerely thank the reviewer for these excellent and highly relevant pointers to related work. We agree that discussing these papers is crucial for positioning our contributions accurately.  We have added this discussion to the revised **Related Work (§6)** to clarify the distinctions.
>
> Regarding the planning literature, we would like to highlight the different goals of our work compared to the papers mentioned: [1] provides a critical benchmark to evaluate if past LLMs can autonomously generate sound and simple plans, finding that their *standalone ability is quite limited*. [2] directly leverages planning to design a new agentic architecture that improves LLM performance.
>
> Our task decomposition experiment (§4.3) has a different purpose. We first claim that "planning improves reasoning", and also show it as a tool to probe the **intrinsic metacognitive abilities** of LRMs. The huge performance gain we show in the multi-turn setting is our primary evidence that LRMs' intrinsic ability to plan is underdeveloped and must be explicitly guided, which directly supports our paper's method design.
>
> [1] On the Planning Abilities of Large Language Models (A Critical Investigation with a Proposed Benchmark). arXiv:2302.06706
> [2] A brain-inspired agentic architecture to improve planning with LLMs. Nature Communication
>
> ***
>
> >**W5:  Addition of Discussion and Limitations**
>
> We are grateful for this suggestion. We agree that a dedicated discussion of limitations and broader implications was a missing piece in our original draft. In our revision, we have significantly expanded the final section (now §7) to include a thorough **Discussion** of the broader context of our work and a new **Limitations**, where we address potential constraints and outline promising directions for future research.

---

> ### Author Response · Authors · 2025-11-28
> **Response to Reviewer DtKq**
>
> Dear Reviewer DtKq,
>
> I hope this message finds you well. As the discussion period is nearing its end. I wanted to ensure we have addressed all your concerns satisfactorily. If there are any additional points or feedback you'd like us to consider, please let us know. Your insights are invaluable to us, and we're eager to address any remaining issues to improve our work.
>
> Thank you for your time and effort in reviewing our paper.
>
> Best regards,
>
> Authors

---

### Official Review · Reviewer_G22W · 2025-10-31

**Soundness:** 3
**Presentation:** 3
**Contribution:** 3
**Rating:** 8
**Confidence:** 3

**Summary:**

Existing Large Reasoning Models (LRMs) may exhibit fragile behaviors in their chain-of-thought reasoning, such as reasoning rigidity or illusory self-correction. The authors argue that if this deficit can be explained as an absence of meta-cognition, and ask whether current LRMs exhibit any analogous capabilities to those in meta-cognition.

**Strengths:**

At first I was skeptical of this paper's claim of positioning metacognition with LRM but I believe by page 7 I became more convinced after seeing the excellent results (e.g., Table 2, Table 3). The paper's motivation, intuition, problem statement, and organization are professionally done and acceptable for ICLR standards. Quality of the figures is also excellent. By far the best part of the paper is section 5. I wish more of the paper was like Section 5, but am also unsure how to edit it to accommodate that. I also appreciate the delicate and grounded approach the authors took in connecting LRMs with metacognition. I strongly urge the authors to review the identified weaknesses in my review as well as the questions I have asked, and see how they might revise this submission to improve upon it.

**Weaknesses:**

"we probe open-source LRMs to determine whether internal computational signals correlate with reasoning outcomes (§3). Our analysis yields a striking finding: signals spanning the entire Transformer architecture—from input-layer attributions to final-layer token probabilities—are highly predictive of answer success. Critically, we demonstrate that correct and incorrect reasoning traces generate statistically distinguishable internal signatures, providing the first empirical evidence that a machine-readable basis for metacognitive
experience exists within these models." <-- isn't this an obvious finding? It is a neural architecture that generates answers. So, of course, there would be intermediate calculations (latent representations) that indicate inevitable success as those intermediate calculations are used to inevitably produce the successful answers. Also, is this really a metacognitive experience? This sort of behavior can be found in the most basic of neural architectures like multi-layer perceptrons if I am understanding this claimed finding correctly.

Line 86-87 "desgining systems that not only..." should be "designing systems that not only..." and "...but cna also act..." should be "...but can also act..."

* What were the t-SNE hyperparameters (e.g., perplexity, iterations) for Figure 2?

* How many epochs were the linear classifiers (probers) trained for on these internal signals as features for predicting final correctness? Was the output actually a correct/fail value? If so, wouldn't it be more appropriate to use a logistic regression or similar? Table 1 AUC seem rather low to claim "...the probers can predict the final correctness of a reasoning trace with an AUC score significantly above chance". IF AUC = 0.5 is random guessing, then AIME2024 w/ Softmax probabilities, Fully-connected activations, and Integrated Gradients are the same or worse than random guessing. Same issue with AIME2025, but also potentially including self-attention scores now. "These results establish that the signals are both distinct and highly predictive, thereby validating our initial hypothesis." I disagree based on the AUC scores. Also, why no formulas or stats on the linear probers (e.g., p-value, R^2) or statistical investigation on if a linear relationship holds?

* Figure 2: The t-SNE of the internal signals do not appear to show different distributions to me. The data looks very embedded, but might only show a slight difference due to alpha of the plotted dots being not exactly 1 or overlap.

* "These token-level scores are then aggregated to produce a trace-level metric, termed of average trace confidence, for each complete solution." (lines 251-252) <- I am assuming arithmetic mean. Should probably avoid the use of "average" as it is less precise. However, the mean alone for describing trace confidence loses a lot of information about the distribution of confidences as its susceptible to outliers. Why not incorporate the standard deviation and other metrics into this analysis when capturing confidence dynamics? Did you investigate if outliers were present? Figure 4 bar plot: are those error bars standard deviation, standard error, or confidence interval?

I find the results of 4.2 for metacognitive monitoring to be lackluster. I am not convinced by Figure 4. And also, discretizing quantitative values into discrete categories (e.g., all confidence above/below a high threshold) although common practice is a bit of a misleading statistical analysis to exaggerate differences. Better practice is to directly analyze the quantitative data with no binning. Also, what are these high thresholds (line 255)?

GRPO algorithm (line 410): write out before using acronym.

Conclusion: I would actually suggest revising this to say something more along the lines that the "...authors empirically demonstrate that while LRMs possess internal signals that *may* be predictive of reasoning outcomes, there is also variability in their usefulness across other domains (e.g., AIME2024, AIME2025); thus, they alone do consistently translate into effective monitoring or control behaviors. To address this gap, ..." Obviously word this better than I have, but I think that performance discrepancy we see on AIME2024 and AIME2025 can actually be addressed in such a way to support your overall proposed method!

**Questions:**

* What were the t-SNE hyperparameters (e.g., perplexity, iterations) for Figure 2?
* How many epochs were the linear classifiers (probers) trained for on these internal signals as features for predicting final correctness? Was the output actually a correct/fail value? If so, wouldn't it be more appropriate to use a logistic regression or similar? Table 1 AUC seem rather low to claim "...the probers can predict the final correctness of a reasoning trace with an AUC score significantly above chance". IF AUC = 0.5 is random guessing, then AIME2024 w/ Softmax probabilities, Fully-connected activations, and Integrated Gradients are the same or worse than random guessing. Same issue with AIME2025, but also potentially including self-attention scores now. "These results establish that the signals are both distinct and highly predictive, thereby validating our initial hypothesis." I disagree based on the AUC scores. Also, why no formulas or stats on the linear probers (e.g., p-value, R^2) or statistical investigation on if a linear relationship holds?

* "These token-level scores are then aggregated to produce a trace-level metric, termed of average trace confidence, for each complete solution." (lines 251-252) <- I am assuming arithmetic mean. Should probably avoid the use of "average" as it is less precise. However, the mean alone for describing trace confidence loses a lot of information about the distribution of confidences as its susceptible to outliers. Why not incorporate the standard deviation and other metrics into this analysis when capturing confidence dynamics? Did you investigate if outliers were present? Figure 4 bar plot: are those error bars standard deviation, standard error, or confidence interval?

* "The loop terminates when sufficient consistency is achieved (e.g., 5 consecutive verification passes) or persistent failure occurs (e.g., a 10-step failure streak)" <- How often did each of this termination conditions occur; what is the ratio?

* "...where $\mathcal{C}$ is a predefined confidence threshold." How is that determined? Hyperparameter?

* Also, what are these high thresholds (line 255)?

---

> ### Author Response · Authors · 2025-11-24
> **Response to Reviewer G22W (1/4)**
>
> Thanks for your valuable feedback and suggestions. We have revised our paper based on your feedback. We detail our response below and please kindly let us know if our response addresses your concerns.
>
> ***
>
> >**W1:  The finding in $\S 3$  is obvious.**
>
> While we agree that a deterministic model's internal states generated the output as it is essentially a mapping process and therefore obvious, we respectfully argue that the internal state does not necessarily guarantee **correctness**. It is **not** obvious that "correctness" (i.e., a complex, semantic outcome often determined only at the end of a reasoning chain) is linearly separable from these internal states. For example, if it is obvious, the phenomenon like "hallucination" (where LRMs generate incorrect answers with high confidence/logits) , can be easily solved by simply looking at the internal states.
>
> Therefore, our finding in $\S 3$ is to reveal a critical dissociation: models possess an internal *signature of error* (distinguishable latent signals) yet fail to manifest this as behavioral self-correction (as shown in $\S 4$).
>
> In the additional experiments, we also proved that this separation phenomenon is affected by the layer number. The results in $\S 5.2$ were based on the self-attention and fully-connected activations signal. We investigated how the performance would change as a result of a change in layer number.
>
> **Table 1: Performance (Self-attention and Fully-connected activation) per layer.**
>
> | layer                          | 1      | 8      | 16     | 24     | 32     | 40     | 48     | 56     | 64     |
> | ------------------------------ | ------ | ------ | ------ | ------ | ------ | ------ | ------ | ------ | ------ |
> | **Self-attention**             | $0.53$ | $0.58$ | $0.63$ | $0.67$ | $0.70$ | $0.70$ | $0.71$ | $0.73$ | $0.73$ |
> | **Fully-connected activation** | $0.55$ | $0.62$ | $0.64$ | $0.66$ | $0.65$ | $0.69$ | $0.69$ | $0.67$ | $0.73$ |
>
> Results in Table 1 show that the performance of our classifier improves with depth of the model layers. Early layers do only slightly better than random chance, while later layers shows significant improvement beyond random chance.
>
> ***
>
> >**W2:  Is this really a metacognitive experience? Similar behavior exists in MLPs.**
>
> We thank the reviewer for raising this concern!  We would like to declare that "**Representation**" (internal embeddings of MLP, often a `direct mapping of features`) is not equal to "**Metacognitive Experience**" (internal signals we discovered is a `dynamic monitoring of processes`, can indicate the states of current reasoning trace).
>
> There is a fundamental distinction between these a simple MLP’s hidden states and the dynamic internal information that can directly experience during the reasoning process:
> + **Different representation objects**: standard MLP's hidden states usually represent the features of the input data, which are typically aligned with the output or corresponding category. However, our analysis on CoT-based LRMs reveals that internal signals track the validity of the reasoning trajectory itself, which is a representation of "the cognitive process itself," not merely a representation of "input data".
> + **Different functions**: instead of simple feature extraction in standard MLPs, the internal signals in LRMs can be used to predict errors even when the mode's explicit output claims certainty. Namely, they can be considered as a functional analogue to a `feeling of error`. Taking "hallucination" as an example, model generates incorrect answers with high confidence. In this case, we can use internal signals as an additional supervision to tackle this miscalibration.
>
>
> ***
>
> >**W3: What is t-SNE settings and visualization is not obvious.**
>
> We apologize for this confusion and have revised the paper to give more details to address these concerns.
>
> **(1) Hyperparameter Details**
>
> For all T-SNE visualization experiments, we use the following settings:
> *   An initial dimensionality reduction to 80 components using PCA (a standard approach to improve t-SNE stability and performance).
> *   A **perplexity of 50** and **1000 iterations**.
>
> **(2) Visualization Results**
>
> Initially, we choose a lower alpha value to mitigate heavy over-plotting, which can provide a more honest sense of data density where correct and incorrect traces overlap. However, we understand the reviewer's point that this can make the outcome confused. To provide full transparency, we have added a new version of the t-SNE plots with `alpha=1.0` in **Appendix A.1**.

---

> ### Author Response · Authors · 2025-11-24
> **Response to Reviewer G22W (2/4)**
>
> ***
>
> >**W4:  Lack of linear classifiers details and statistical analysis.**
>
> Very thanks for for this insightful question. We apologize for the confused words and have revised $\S 3$ and  $\S 7$ for clarity.
>
> **(1) Clarification on Prober Method:**
>
> Here, this is a binary classification task (predicting a correct/fail outcome), and our "linear classifier" is indeed a **logistic regression**, which is the standard and appropriate method. We apologize for the ambiguous words and have make it more clear in $\S 3$ .
>
> For AIME datasets evaluation, the logistic regression prober were trained for 10 iterations on a mix of GSM8K and MATH datasets. Due to the very small size of the AIME dataset (n=30), we used it exclusively as a test set for out-of-distribution evaluation. we have also revised the $\S 3$ and added the training details to **Appendix A.1**.
>
> **(2) Revise Results on AIME dataset**
>
> We agree that our original claim that the signals were "highly predictive" across all benchmarks was not concise enough.  Actually, when the prober from easy benchmark is evaluated on a more challenging and hard benchmark like AIME, its performance drops to near-random chance. Through this observation, we conclude that **the predictive power of an LRM's innate metacognitive signals appears brittle and task-dependent.** I think it is somehow interesting since the results align with the concept of **metacognitive experience**: dynamic computational signals. Different tasks may exhibit different pattern signals from representation space.
>
> Here very thanks for your suggestions (in conclusion) to help us revise our paper! We have revised the saying in $\S 7$ .
>
> **(3) Statistical Analysis**
>
> Here we conducted a detailed analysis of our logistic regression prober on the GSM8K dataset, and the results are shown in Table 2.
>
> **Table 2: R-squ and p-value analysis.**
>
> | Internal Signal Source    | Pseudo R-squ. | p-values (x) | p-values (y) |
> | :------------------------ | :------------ | :----------- | :----------- |
> | **Self-attention scores** | **$0.2823$**  | $0.029$      | $0.017$      |
> | **Fully activation**      | **$0.2603$**  | $0.014$      | $0.003$      |
> | Softmax probabilities     | $0.1477$      | $0.013$      | $0.055$      |
> | Integrated Gradient       | $0.0141$      | $0.439$      | $0.246$      |
> *(p-values < 0.05 indicate statistical significance)*
>
> The analysis reveals that signals from **Self-attention** and **Fully activation** have a statistically significant relationship with reasoning correctness (p < 0.05). However, the modest Pseudo R-squared values suggest that a linear model can only capture a fraction of the predictive information. So I think the relationship between internal signals and correctness is likely *more complex*. Using a more powerful non-linear prober like MLP would be stronger to explore the full extent of this separability. We have integrated discussion and details in **Appendix A.1** and thank you again for pushing us to strengthen this crucial part of our work.

---

> ### Author Response · Authors · 2025-11-24
> **Response to Reviewer G22W (3/4)**
>
> ***
>
> >**W5:  Confidence Metrics and Results in $\S 4.2$.**
>
> We thank the reviewer for this statistical suggestions for our confidence define. Here we provide statistical analysis for this part.
>
> **(1) Definitions**
>
> Our *average trace confidence* indeed refers to the **arithmetic mean** of token-level confidences. Next, we clarify that the *error bars* represent the **Standard Deviation (SD)**, illustrating the variability of confidence scores across the sample population. Also, the thresholds for *"High/Low" confidence were defined dynamically as the **Top 15%** and **Bottom 15%** of the LRM's confidence distribution, respectively.
>
> **(2) Distributional Analysis**
>
> The mean alone may mask the internal dynamics of the confidence distribution. To address this, we analyzed the SD of confidence scores.  Incorrect traces exhibit **higher internal deviation** ($0.943$) compared to correct traces ($0.507$).
>
> This nearly $2\times$ difference indicates that incorrect reasoning is intrinsically volatile, fluctuating between high and low certainty. However, correct reasoning is **stable**. This confirms that "distributional stability" is a distinct and predictive metacognitive signal, which validates our motivation to monitor these dynamics.
>
> **(3) Outlier Effect**
>
> To address this issue, we added statistical analyses through **Median** and **Lowest 10% Group** metrics on the GSM8K dataset to show robustness. Here, *Median* means we choose median token confidence of entire reasoning traces. To further consider outlier effect, we first divided the reasoning traces into adjacent groups (here we choose $512$ tokens), then we calculate the mean of the bottom 10% of group confidences within the trace. Such *Lowest 10% Group* design could capture localized and smoother signal. Here are some results.
>
> **Table 3: Comparison of Confidence Aggregation Metrics (Qwen3-32B).**
>
> | Metric               | Correct Mean | Incorrect Mean |
> | :------------------- | :----------: | :------------: |
> | **Mean** (Original)  |   $17.17$    |    $14.92$     |
> | **Median**           |   $15.91$    |    $12.32$     |
> | **Lowest 10% Group** |   $14.22$    |    $10.48$     |
>
> As shown in Table 3,  this three metric shows the exact same trend, confirming that our observation is **not heavily driven by outliers**. For *Lowest 10% Group* which focuses on the "weakest links" in reasoning. The larger difference shows that: In incorrect samples, outliers (e.g., high confidence) improve the *Mean* score. By remove them away (*Median/Lowest 10%*), the distinction becomes sharper. Thus, our original finding based on Mean was a lower-bound estimate of the model's metacognitive capability.
>
> **(4)  Revise Results in $\S 4.2$**
>
> Actually, we tried before to analyze the **continuous confidence dynamics** using the slope $\Delta = C_{\text{final}} - C_{\text{start}}$. For Qwen3-32B, the mean of on incorrect samples is $-0.417$, with $37.2$% of incorrect traces showing a positive slope. The statistical characteristics are not obvious, which somehow shows weak confidence calibration.
>
> However, such linear random variable fails to capture the **non-linear nature of reasoning**.
> As reasoning involves backtracking and sub-goal verification, confidence trajectories often exhibit complex shapes (e.g., U-shaped).  So we focus on identifying specific metacognitive profiles (e.g., "Overconfidence", "Over-cautious") that global statistics miss. Here we agree with the reviewer's concern that discrete settings may risk exaggerating differences. To avoid over-generalization, we have revised the **results** in $\S 4.2$ to clarify that the "pattern-based" metric applies to identifying specific misalignment behaviors rather than describing the entire distribution.  We replaced strong claims like *"confidence often increases"* with more precise descriptions of inconsistency and misalignment. This accurately reflects that while not all incorrect traces show increasing confidence, a significant portion does, proving that the model's confidence is an unreliable predictor of correctness.
>
> We have revised the paper and put this details in **$\S 4.2$** and **Appendix.A.1**

---

> ### Author Response · Authors · 2025-11-24
> **Response to Reviewer G22W (4/4)**
>
> ***
>
> >**W6: Typo and format error.**
>
> Very thanks for the reviewer to pointing out this typo, and we have revised this typo in $\S 1$. Next, we have write out *Group Relative Policy Optimization* before using acronym in $\S 5.2$.
>
> ***
>
> >**Q1: The metacognitive loop details.**
>
> To answer this question, we give our termination logs for DeepSeek-R1 on the AIME 2024 and AIME 2025 datasets (60 problems total). The termination statistics are summarized below:
>
> **Table: Frequency of Termination Conditions**
>
> |           |  1   |  2  |  3  |  4  | 5   | 6   | 7   | 8   | 9   | failed |
> | :-------- | :--: | :-: | :-: | :-: | --- | --- | --- | --- | --- | ------ |
> | AIME 2024 | $18$ | $6$ | $0$ | $2$ | $1$ | $0$ | $2$ | $0$ | $0$ | $1$    |
> | AIME 2025 | $15$ | $3$ | $1$ | $2$ | $1$ | $0$ | $1$ | $3$ | $1$ | $3$    |
>
> We can see 30% (9/30) of successes required adaptive adjustments in 2-5 round loops on AIME2024. Also, 40% of successes required more than 1 round loops on AIME2025, which demonstrates that the metacognitive loop effectively recovers from initial uncertainty or errors in hard problems.
>
> ***
>
> >**Q2: Clarification on the confidence threshold $\mathcal{C}$ determination.**
>
> To set the threshold $\mathcal{C}$, we calculate the bottom $\gamma$-th percentile ($\gamma=10\%$) of the token confidence observed from the cold-start SFT model on a validation set from MATH. In our experiment, we here calculate the threshold $\mathcal{C}=13.2$ based on the hyperparameter $\gamma=10\%$.

---

> ### Author Response · Authors · 2025-11-28
> **Response to Reviewer G22W**
>
> Dear Reviewer G22W,
>
> I hope this message finds you well. As the discussion period is **nearing its end**. I wanted to ensure we have addressed all your concerns satisfactorily. If there are any additional points or feedback you'd like us to consider, please let us know. Your insights are invaluable to us, and we're eager to address any remaining issues to improve our work.
>
> Thank you for your time and effort in reviewing our paper.
>
> Best regards,
>
> Authors

---

### Official Review · Reviewer_fMP3 · 2025-11-01

**Soundness:** 3
**Presentation:** 3
**Contribution:** 2
**Rating:** 4
**Confidence:** 4

**Summary:**

The paper proposes a functional framework for metacognition in Large Reasoning Models (LRMs), separating information from abilities. It first shows that internal signals (final-layer probs, activations, attention, IG) predict success, then evaluates monitoring and control. To bridge them, it introduces (i) prompt-based and (ii) fine-tuning based methods.

**Strengths:**

Clear functional framing (information vs abilities) and a helpful suite of behavioral diagnostics; writing and figures are generally clear.

**Weaknesses:**

While the framing is tidy, the methods largely mirror existing practice:

- In-context scaffolding / multi-step prompting resembles established prompting frameworks (self-consistency, plan-and-solve, Tree-of-Thoughts, debate/verification variants).
- Fine-tuning on reasoning/process traces closely relates to STaR and subsequent process-supervision works.
- Predicting success from internal signals (log-probs/confidence/activations) is well-documented (“LMs mostly know what they know”).

Overall I view the conceptual clarity as good, but the technical originality as moderate: the core ingredients—prompting, self-verification, process-supervised fine-tuning, and internal confidence probes—are known.

I think the paper would benefit from clarifying what is new beyond re-packaging.

**Questions:**

See above.

---

> ### Author Response · Authors · 2025-11-23
> **Response to Reviewer fMP3 (1/4)**
>
> We thank reviewer fMP3 for providing valuable suggestions. We have revised our paper based on your feedback. We detail our response below and please kindly let us know if our response addresses your concerns.
>
> ***
>
> >**W1:  Differentiation from Existing Prompting Strategies and Method Novelty.**
>
> We would like to clarify the difference between existing practices and our emergent metacognitive reasoning system.
>
> In §5.1, our goal is to construct the first **Externalized Metacognitive Reasoning Paradigm**. While prior works only address specific sub-problems of reasoning, our framework orchestrates them into an **end-to-end Metacognitive Loop (Monitoring $\leftrightarrow$ Control)**, enabling dynamic regulation that static prompting pipelines cannot achieve.
>
> **(1) Differences between our system and existing works**
> First, we analyze the specific differences between our system and the baselines mentioned [1-4]:
>
> [1] **Self-Consistency** is a decoding strategy that relies on sampling diverse paths and aggregating final answers (**outcome-focused**). It lacks the ability to intervene during the reasoning process. In contrast, our system implements **process-focused control**: it monitors confidence in real-time and actively refine the reasoning trajectory before the final answer is reached.
>
> [2] **Plan-and-Solve** utilizes a **static pipeline** (Plan $\rightarrow$ Execute). It cannot shift strategy if the initial plan is flawed. Our system employs **dynamic control**: if the *Confidence Monitor* detects high uncertainty during execution, the *Strategy Regulator* interrupts the process to revise the plan.
>
> [3] **Tree of Thoughts (ToT)** is a heuristic search algorithm (e.g., BFS/DFS) to explore multiple reasoning paths based on predefined hyperparameters (width/depth). Our framework leverages **internal uncertainty signals** to determine when to branch out and which strategy to employ.
>
> [4] **Self-Verification** adds a **post-hoc** backward verification process (using the final answer to deduce initial conditions) to rank candidate solutions after generation is complete. Our framework **online monitoring**: it assesses the reasoning process in real-time and shifts the paradigm from outcome verification to process regulation.
>
> **(2) Ablation Study
>
> To prove that our performance gains stem from the collaboration of the metacognitive loop rather than individual prompting tricks, we conducted a comprehensive **Ablation Study** on the AIME and GPQA benchmark using DeepSeek-R1.
>
> **Table 1: Ablation Study of Metacognitive Components (AIME 2025，GPQA)**
>
> | Method                        | Pass@1(AIME) | $\Delta$  | Pass@1(GPQA) |  $\Delta$  |
> | :---------------------------- | :----------: | :-------: | :----------: | :--------: |
> | *Baseline (Vanilla CoT)*      |    87.5%     |           |    81.0%     |            |
> | *(a) w/o Planning.*           |    89.3%     |   +1.8%   |    90.2%     |   +9.2%    |
> | *(b) w/o Confidence Monitor*  |    88.9%     |   +0.6%   |    87.5%     |   +6.5%    |
> | *(c) w/o Strategy Control*    |    88.1%     |   +1.4%   |    85.3%     |   +4.3%    |
> | **Full Metacognitive System** |  **90.0%**   | **+2.5%** |  **93.4%**   | **+12.4%** |
>
> As shown in Table 1, removing the strategy control causes the largest performance drop. This confirms that flexibility is important. Monitoring is also important, the system tends to loses the trigger for self-correction. The results shows that our metacognitive loop is critical.

---

> ### Author Response · Authors · 2025-11-23
> **Response to Reviewer fMP3 (2/4)**
>
> >**W1:  Differentiation from Existing Prompting Strategies and Method Novelty. (Continued)**
>
> **(3) Open Discussion: From Eliciting Metacognition to Optimizing It**
>
> We appreciate the reviewer's perspective, as it touches upon an important point in developing metacognitive system. Here I want to discuss and share the thought process behind our approach in Section 5.
>
> Our idea for the **emergent metacognitive system** comes from the prior observations. For instance, in §4.3, we observed that simply separating the task into a 'plan' and a 'solve' step already improved performance. This led us to wonder: *what if we built a more complete loop where different roles—a planner, a solver, a monitor, and a controller—could work together?*
>
> The ablation study above confirms that the performance gains are not merely additive effects of prompting tricks, but rather stem from the **collaboration** within the complete metacognitive loop (Monitoring ↔ Control). This emergent system, where modular roles are orchestrated to self-regulate the reasoning process, is the core contribution of our prompt-driven framework.
>
> However, we guess that this is just the first step. Our findings in §4 reveal that the inherent metacognitive abilities of the underlying LRM like self-awareness can be fragile. This raises a critical question for the field: **Can we move beyond simply eliciting these capabilities in-context and begin to explicitly train and optimize the collaborative dynamics of the entire system?**
>
> To this end, we have initiated an exploration into this very direction. We propose a novel RL framework, **MeRSL (Metacognitive Reasoning System Learning)**, designed to strengthen the connections within the metacognitive loop. Our **two-stage** approach first optimizes the foundational "plan-and-solve" hierarchy to establish structured forward reasoning. Subsequently, it fine-tunes the adaptive "monitor-and-correct" dynamics to enable robust self-correction. Our preliminary results on `Qwen2.5-7B-Instruct` are as follows:
>
> **Table: Stage-wise MeRSL results with a single correction round at inference.**
>
> | Dataset   | Prompt-based Loop | Stage 1 | Stage 2 | Stage 1 + Stage 2 |
> | :-------- | :---------------: | :-----: | :-----: | :---------------: |
> | `MATH500` |       75.3%       |  76.2%  |  77.2%  |     **79.4%**     |
> | `GSM8K`   |       92.2%       |  91.8%  |  92.6%  |     **92.6%**     |
>
> We have included a detailed method and results in **Appendix A.6**. We believe this exploration opens up a new thinking for metacognitive system research, paving the way for future work.
>
> **References:**
>
> [1]. *Self-Consistency Improves Chain of Thought Reasoning in Language Models.* ICLR 2023
>
> [2]. *Plan-and-Solve Prompting: Improving Zero-Shot Chain-of-Thought Reasoning by Large Language Models.* ACL 2023
>
> [3]. *Tree of Thoughts: Deliberate Problem Solving with Large Language Models.* Nips 2023
>
> [4]. *Large Language Models are Better Reasoners with Self-Verification*, EMNLP 2023

---

> ### Author Response · Authors · 2025-11-23
> **Response to Reviewer fMP3 (3/4)**
>
> ***
>
> >**W2:  Comparison with STaR and our fine-tuning method.**
>
> We thank the reviewer for highlighting the connection to STaR [5]. Here we give a comparison with STaR and our fine-tuning method.
>
> While we share the broad methodology of fine-tuning, our work diverges from STaR in **data structure**, **cognitive level**, and **training paradigm**.
>
> **(1) Object-level vs Meta-level**
> STaR **(Object-level)** focuses on bootstrapping the correctness of the reasoning chain itself (i.e., `Question` $\rightarrow$ `Rationale (CoT)` $\rightarrow$ `Answer`). Ours **(Meta-level)** approach models the **metacognitive processes** that conduct planning and self-awareness. Our training data introduces structured tags that do not exist in STaR:
>
> -   `<difficulty>...</difficulty>`: to perform **self-assessment** before reasoning.
> *   `<plan>...</plan>`: to perform **high-level task decomposition**.
> *   `<think>...</think>`: reasoning (which maps to STaR's rationale).
>
> By fine-tuning these meta-components, we train the model not just *how to solve* (reasoning), but also *how to plan* (control) and *how to assess* (monitoring) a problem.
>
> **(2) Training Paradigm**
> STaR relies on a self-training loop where the model learns from its own correct outputs to improve performance iteratively. In contrast, our **Cold-start SFT** injects **structured metacognitive priors** (derived from expert models like Gemini-2.5-Pro) that the base model initially lacks. We are not only filtering correct trajectories, but also teaching a new cognitive pattern (Assess $\rightarrow$ Plan $\rightarrow$ Solve).
>
> **(3) Complementary Nature**
> In addition, our fine-tuning method can also be viewed as an orthogonal extension to STaR-like methods. While STaR improves the quality of the `<think>` process, our framework ensures that this thinking process is grounded in accurate self-awareness and structured planning.
>
> To prove that, we conducted a controlled experiment combining our Metacognitive Schema with the STaR iterative loop. We run on MATH dataset using Qwen2.5-Math-7B as the base model:
>
> * **STaR:** Iteratively fine-tuning on standard `Q -> CoT -> Answer` data.
> * **Meta-STaR (Ours):** Iteratively fine-tuning on `Q -> <difficulty> -> <plan> -> CoT -> Answer` data.
>
> Here are the results:
>
> **Table 3: Performance on Math-500 between STaR and Ours**
>
> | Method               | Iteration 0 | Iteration 2 | Iteration 4 |
> | :------------------- | :---------: | :---------: | ----------- |
> | Qwen2.5-Math-7B      |    64.0%    |      -      | -           |
> | **STaR**             |    64.0%    |    65.7%    | 67.0%       |
> | **Meta-STaR (Ours)** |    64.0%    |    68.6%    | 71.2%       |
>
> As shown in Table 3, we can notice that applying the our metacognitive to STaR algorithm fine-tuning design can lead to higher performance than standard STaR.  We have revised the paper and put these details in **Appendix.A.7**.
>
> **References:**
>
> [5] *STaR: Self-Taught Reasoner Bootstrapping Reasoning With Reasoning*. NIPS 2022.

---

> ### Author Response · Authors · 2025-11-23
> **Response to Reviewer fMP3 (4/4)**
>
> ***
>
> >**W3:  Predicting Success from Internal Signals is Well-Documented.**
>
> We thank the reviewer for referencing [6]. We fully agree that the predictive power of internal signals is well-established (also discussed in §3). However, as we mentioned, it is hard for current LRMs to leverage this information as observable metacognitive abilities.
>
> So in §5, our contribution is not re-discovering this correlation, but **internalizing it within a Reinforcement Learning framework** to bridge this information-to-ability gap.
>
> **(1) From Passive Observation to Active Control**
> Prior work [6] focuses on measuring calibration or selecting best answers post-hoc. They establish that the signal exists.
>
> Based on this, we treat this signal as a **control signal for exploration** and propose our RL algorithm. In our algorithm, low confidence does not just signal *“might be wrong"*; it triggers a **dynamic fork** in the reasoning tree (line 425-430), exploring more rollouts strategies to uncertain steps during training. This transforms a passive metric into an active training mechanism.
>
> **(2) Robustness to Signal Choice**
> To further prove that our novelty lies in the algorithm (monitoring+strategy control) rather than the specific choice of signal, we conducted an ablation study comparing different confidence metrics within our RL framework on the MATH500 dataset.
>
> **Table 4: RL Performance with different monitoring signals.** The random setting is to fork randomly based on confidence until the same group size is reached.
>
> | Metric Type                 | MATH500 (Pass@1) | GSM8K (Pass@1) |
> | :-------------------------- | :--------------: | :------------: |
> | Qwen2.5-Math-7B             |      64.0%       |     70.3%      |
> | *Baseline (GRPO)*           |      71.6%       |     75.9%      |
> | Random                      |      66.3%       |     71.4%      |
> | Sentence-level Confidence   |      72.4%       |     79.1%      |
> | Sentence-level Entropy      |      73.0%       |     78.8%      |
> | **Token Confidence (Ours)** |    **80.2%**     |   **85.5%**    |
>
> As shown in Table 4, we can notice that the choice of metric like sentence entropy and confidence yields small performance difference, yet all outperform the baseline GRPO. Meanwhile, a random control setting brings performance degradation to GRPO. This confirms that the performance gain mainly stems from the **metacognitive control mechanism** (allocating exploration where it is needed).  We have revised the paper and put these details in **Appendix.A.7**.
>
> **Reference:**
>
> [6] *Language Models (Mostly) Know What They Know*.  arXiv:2207.05221, 2022.

---

> ### Author Response · Authors · 2025-11-28
> **Response to Reviewer fMP3**
>
> Dear Reviewer fMP3,
>
> I hope this message finds you well. As the discussion period is **nearing its end**. I wanted to ensure we have addressed all your concerns satisfactorily. If there are any additional points or feedback you'd like us to consider, please let us know. Your insights are invaluable to us, and we're eager to address any remaining issues to improve our work.
>
> Thank you for your time and effort in reviewing our paper.
>
> Best regards,
>
> Authors

---

### Official Review · Reviewer_TDuK · 2025-11-01

**Soundness:** 3
**Presentation:** 3
**Contribution:** 2
**Rating:** 4
**Confidence:** 4

**Summary:**

This paper proposes two methods to understand the metacognition; (i) prompt-guided role-playing. It is to assign distinct metacognitive roles—such as ‘Planner’, ‘Solver’, and ‘Verifier’—to simulate a complete monitoring and control loop. (ii) enriches model’s
metacognitive knowledge through fine-tuning. They construct a dataset with explicit metacognitive annotations (e.g., plans, self-corrections) and fine-tune the model using a hybrid learning objective, directly embedding these capabilities into its parameters.

Specially, they
- evaluate the metacognition on a series annotated tasks, i.e., self-awareness, difficulty, task decomposition and flexibility.  based on annotated dataset, for example, difficulty-level dataset, then prompt the model to assess the  difficulty of the input task and record their performance as ACC.
- based on the observation that LLM fail to exhibit desirable metacognition, they use role-playing method to elicit their internal awareness
- they further improve LLM's metacognition by training LLM on annotated metacognition-aware trajectories.

**Strengths:**

1. It studies the metacognition in knowledge and experience, two aspects, and design four tasks to assess the LLM's capabilities.
2. It proposes two methods to enhance model's metacognition, both prompt-guided role playing and fine-tuning.

**Weaknesses:**

1. In section 4, the author design 4 tasks to evaluate the model's metacoginition,
- while some of these experiments are already evaluated in existing literature, such as difficulty-level, and self-awareness.  "Do Large Language Models Know What They Don't Know?"
- the experiment design is not rigorous. for example, the task decomposition. It shows that decomposition can improve performance, are you evaluating for the same question, with/without decomposition? Also, it is similar to Chain-of-thought setting, which has already shown that with step-by-step thinking, the model can think better.
2. Both prompt-guided role and fine-tuning methods are not very insightful.
- for prompt-based method, what is your vanilla setup? direct prompt? but your proposed method actually incorporate self-reflection loop, which has been shown to be effective than direct prompt or even CoT. so the performance boots for me is not very insightful.
- for fine-tuning methods, do you have any OOD evaluation? and what is your prompt in Qwen2.5-Math-7B when inference?

**Questions:**

See questions above.

---

> ### Author Response · Authors · 2025-11-23
> **Response to Reviewer TDuK (1/3)**
>
> We thank reviewer TDuK for providing positive feedback and helpful suggestions. We have revised our paper based on your feedback. We detail our response below and please kindly let us know if our response addresses your concerns.
>
> ***
>
> >**W1:  Clarification between our Self-Awareness and Existing Literature.**
>
> We thank the reviewer for referencing the important work "Do Large Language Models Know What They Don't Know?" [1] We appreciate the opportunity to clarify the difference between **self-knowledge** [1] and **difficulty assessment** (§4.1), and to present new experiments expanding our evaluation scope.
>
> **(1) Concept Difference**
> While the existing study [1] primarily evaluates whether LLMs can identify unanswerable questions (static knowledge boundaries), our "Difficulty Assessment" task in §4.1 targets whether LRMs can assess a question’s intrinsic complexity. Such abilities is important for **estimating the cognitive load required for reasoning.**
>
> In **cognitive science**, self-awareness involves recognizing one’s own cognitive states, including having a clear awareness of one’s specific task difficulties and goals [2]. This awareness serves as a trigger to optimize **resource allocation**.
>
> For example, a person or an agent must decide:*"Does this require a complex, multi-step strategy or a simple retrieval?"*. Therefore, difficulty assessment is pivotal for LRM metacognition research, as it acts as a process monitoring signal that directly informs **control strategies** (e.g., *"Should I plan first?"* or *"Should I verify?"*).
>
> Based on this motivation, we mainly focused on difficulty assessment in §4.1 to validate the model's potential for dynamic reasoning control.
>
> **(2) Additional Experiments: LRMs Logic and Knowledge Awareness**
> We acknowledge that the knowledge boundary awareness mentioned by the reviewer is also an integral component of self-awareness. To totally address the reviewer's suggestion and enrich our metacognitive framework, we added two new experiments in **Appendix A.2**:
>
> **Experiment 1: Logic Self-Awareness**
> To test if LRMs are aware of their own reasoning logic, we constructed a causal masking task. We paired questions from DeepMath with their correct CoT and masked crucial logical connectors using Gemini-2.5-Pro (e.g., *So*, *But*, *Alternatively*). The model was asked to choose the masked logic words.
>
> **Table 1: Accuracy of Predicting Logical Connectors**
>
> | Model              | Overall Acc | "So" (Causal) | "But" (Turn) | "Alternatively" (Branch) |
> | :----------------- | :---------: | :-----------: | :----------: | :----------------------: |
> | **Gemini-2.5-Pro** |    60.9%    |     92.1%     |    55.4%     |          35.2%           |
> | **GPT-OSS-120B**   |    74.7%    |     94.3%     |    71.2%     |          58.6%           |
> | **Qwen3-8B**       |    47.3%    |     88.5%     |    32.3%     |          21.1%           |
>
> As shown in Table 1, LRMs show a strong bias towards causal reasoning ("So") but struggle significantly with branching ("Alternatively") and turning points ("But"). This may reveals models are often unaware the relation between reasoning steps.
>
> **Experiment 2: Knowledge Boundary Awareness**
> To directly address the "Know What They Don't Know" aspect, we modified 200 samples from DeepMath by removing key conditions to make them **Unsolvable**, and mixed them with 200 **Solvable** problems. We asked models to judge: *"Is this problem solvable?"*
>
> **Table 2: Solvable vs. Unsolvable**
>
> | Model              | Solvable Acc | Unsolvable Acc |
> | :----------------- | :----------: | :------------: |
> | **Gemini-2.5-Pro** |    96.5%     |     41.5%      |
> | **GPT-OSS-120B**   |    98.0%     |     58.0%      |
> | **Qwen3-8B**       |    86.5%     |     12.5%      |
>
> As shown in Table 2, We can see LRMs can easily identify solvable problems, they sometimes exhibit slightly overconfidence on unsolvable ones. It suggests that current LRMs still sometimes fail to accurately identify knowledge boundaries. We have revised the paper and put these details in **Appendix.A.2**.
>
> [1] *Do Large Language Models Know What They Don't Know?* ACL 2023
>
> [2] *Metacognition and affect: What can metacognitive experiences tell us about the learning process?*  Educational Research Review 2006

---

> ### Author Response · Authors · 2025-11-23
> **Response to Reviewer TDuK (2/3)**
>
> ***
>
> >**W2: Rigor of Experimental Design and Distinction from CoT.**
>
> We thank the reviewer for this crucial question. We apologize if the description in Figure 5 caused confusion regarding the experimental setup. We are happy to clarify the rigor of our design and the fundamental distinction between our **Task Decomposition (Metacognitive Control)** and **Chain-of-Thought (Cognitive Execution)**.
>
> **(1) Clarification on Experiment Design**
> To ensure a rigorous comparison, our evaluation was conducted on the **same set of questions** across all three conditions. The variables were controlled as follows:
> *   **Baseline (CoT):** The model performs standard CoT (`"Please reason step by step"`).
> *   **Intervention 1 (Planning + CoT):** The model is prompted to generate a plan and execute CoT in one context.
> *   **Intervention 2 (Turn 1: Planning, Turn 2: CoT):** A two-stage process where the model *only* generates a plan first, which is then fed back to guide the CoT reasoning.
>
> **(2) Concept Difference: Planning vs. CoT**
> While both involve generating intermediate tokens, they occupy different levels in the cognitive hierarchy:
> *   **CoT (Cognition):** This is the **Low-level execution** (e.g., calculating an equation, deriving a specific lemma).
> *   **Task Decomposition (Metacognition):** This is the **High-level control** (e.g., "First I need to find X, then use Theorem Y").
>
> Our results support this distinction. As shown in Fig. 5, the **Multi-turn Condition** (where planning is isolated) significantly outperforms the Baseline. This gain suggests that explicitly engaging a planner mode before the solver mode prevents the model from getting lost in calculation details, a benefit distinct from simply "thinking longer."
>
> **(3) Experiment Analysis: Planning vs. CoT**
> Thanks for your valuable suggestions! To further prove that decomposition is a distinct from CoT, here we performed an ablation study between *standard CoT* and *only generating a plan and directly give the answer* on the dataset mentioned in §4.3.
>
> **Table 3: Accuracy between planning and CoT (Qwen2.5-Math-7B).**
>
> | Model              | Task Decomposition<br>(without CoT) | CoT | Task Decomposition+CoT |
> | :----------------- | :---------------------------------: | :-: | ---------------------- |
> | **Gemini-2.5-Pro** |                 34%                 | 34% | 42%                    |
> | **GPT-OSS-120B**   |                 36%                 | 36% | 40%                    |
> | **Seed-1.5-Pro**   |                 41%                 | 46% | 52%                    |
> | **Qwen3-8B**       |                 18%                 | 14% | 26%                    |
> | **Qwen3-32B**      |                 23%                 | 22% | 38%                    |
>
> As shown in Table 3, we can see only task decomposition does not provide a significant performance boost when it exists in isolation; it usually becomes effective when based on a consistent reasoning process (CoT).
>
> We have revised **§4.3** to clarify these definitions and included this stratified analysis in **Appendix A.4** to strengthen the claim.

---

> ### Author Response · Authors · 2025-11-23
> **Response to Reviewer TDuK (3/3)**
>
> ***
>
> >**W3: Clarification on Baseline Fairness and the Insightfulness of the Metacognitive Loop.**
>
> We thank the reviewer for this valuable question. We apologize for not explicitly detailing the "Vanilla" setup in the main text, which may led to the confusion. Here we clarify our settings and also add it in our **Appendix.A.6** .
>
> **(1) "Vanilla" setup**
> To ensure our gains are not merely due to increased computation (more tokens/calls), here our **Vanilla setup is a "Best-of-K" CoT baseline**, not a single-pass direct prompt.
> *   **Setup:** We sample $K$ independent Chain-of-Thought paths for the same question (Prompt: *“please reasoning step by step, state the final answer within \boxed{}”*) and consider it a success if *any* path yields the correct answer (Pass@K).
> *   **Compute Matching:** In our experiment, we set $K=7$, which aligns with the average number of API calls triggered by our Metacognitive System (Upper Bound $\approx 7$ steps).
>
> **(2) Ablation Study**
> To prove that our performance boots stem from the *collaboration* of our dynamic metacognitive loop. Here we conducted a comprehensive ablation study on AIME 2025 and GPQA using DeepSeek-R1.
>
> **Table 4: Ablation Study of Metacognitive Components.**
>
> | Method                         | Pass@1(AIME) | $\Delta$  | Pass@1(GPQA) |  $\Delta$  |
> | :----------------------------- | :----------: | :-------: | :----------: | :--------: |
> | *Baseline (Vanilla Best-of-7)* |    87.5%     |           |    81.0%     |            |
> | *(a) w/o Planning.*            |    89.3%     |   +1.8%   |    90.2%     |   +9.2%    |
> | *(b) w/o Confidence Monitor*   |    88.9%     |   +1.4%   |    87.5%     |   +6.5%    |
> | *(c) w/o Strategy Control*     |    88.1%     |   +0.6%   |    85.3%     |   +4.3%    |
> | **Full Metacognitive System**  |  **90.0%**   | **+2.5%** |  **93.4%**   | **+12.4%** |
>
> Removing the strategy control causes the largest performance drop. This confirms that strategy flexibility is important. Monitoring is also important, the system tends to loses the trigger for self-correction. The results shows that our metacognitive loop paradigm benefits our performance. For more ablation study details, please see **Appendix.A.6**.
>
> ***
> >**W4: OOD Generalization Analysis and Inference Setup.**
>
> We thank the reviewer for inquiring about the generalization capability of our method. We agree that demonstrating performance on Out-of-Distribution (OOD) tasks is crucial to verifying that our **Internalized Metacognition** model has learned a transferable reasoning mechanism. Here are our results.
>
> **(1) OOD Evaluation Results**
> To validate this, we evaluated our fine-tuned model (Ours: SFT+RL) against the Base model (`Qwen2.5-Math-7B`) and the standard RL baseline (`GRPO`) on four diverse benchmarks:
> *   **ARC-c**: Open-domain reasoning.
> *   **GPQA-Diamond**: Graduate-level science (Biology, Physics, Chemistry).
> *   **MMLU-Pro**: Complex reasoning across diverse academic subjects.
> *   **LiveCodeBench**: Code generation.
>
> **Table 5: Performance on Out-of-Distribution Benchmarks.**
>
> | Method                           |  ARC-c   | GPQA-Diamond | MMLU-Pro | LiveCodeBench |
> | :------------------------------- | :------: | :----------: | :------: | :-----------: |
> | **Base Model** (Qwen2.5-Math-7B) |   18.2   |     33.8     |   37.4   |     28.9      |
> | **GRPO** (Standard RL)           |   29.8   |     39.2     |   42.1   |     35.5      |
> | **Ours (SFT+RL)**                | **33.2** |     46.5     |   47.6   |     40.4      |
>
> As shown in Table 5, our method achieves boosts on GPQA and MMLU. This confirms that the internalized metacognitive skills, such as *difficulty assessment*, *planning*, and *monitoring*, are domain-agnostic. On LiveCodeBench, our method shows a modest improvement. We attribute the smaller gain to the base model architecture (`Qwen2.5-Math`), which is specialized for mathematics rather than coding.
>
> **(2) Inference Prompt Details**
> To ensure reproducibility and fair comparison, we used a standard CoT trigger without any task-specific engineering during inference for all models (Base, GRPO, and Ours):
>
> `Please reason step by step and put the final answer within \boxed{}.`
>
> This simple prompt ensures that the performance gains are driven by the model's **intrinsic parameter updates** (internalized metacognition) rather than prompt engineering. We have revised the paper and put these details in **Sec.5.2** and **Appendix.A.7**.

---

> ### Author Response · Authors · 2025-11-28
> **Response to Reviewer TDuK**
>
> Dear Reviewer TDuK,
>
> I hope this message finds you well. As the discussion period is **nearing its end**. I wanted to ensure we have addressed all your concerns satisfactorily. If there are any additional points or feedback you'd like us to consider, please let us know. Your insights are invaluable to us, and we're eager to address any remaining issues to improve our work.
>
> Thank you for your time and effort in reviewing our paper.
>
> Best regards,
>
> Authors

---

### Author Response · Authors · 2025-11-29
**General Response**

Dear AC and Reviewers,

We sincerely thank every reviewer for your time and patience to review our paper and provide insightful feedback. Here, apart from the response to each reviewer, we would like to thank the reviewers for acknowledging our work and highlight new results added during the rebuttal. We are encouraged that the reviewers appreciate and recognize our contributions:

**Novel Framework & Perspective:**
*   **A "delicate and grounded approach"** connecting LRMs with metacognition, with a motivation and problem statement that are **"professionally done"**. [G22W]
*   **"Clear functional framing"** that separates internal information from behavioral abilities, providing conceptual clarity. [fMP3]
*   Investigates metacognition through two distinct aspects: **knowledge and experience**. [TDuK]
* Addresses an **"important and timely"** topic in the reasoning model community. [DtKq]

**Evaluation:**
*   Performs **"a series of empirical evaluations"** with a **"helpful suite of behavioral diagnostics"** to systematically measure metacognitive abilities. [DtKq, fMP3]
*   Designs **four targeted tasks** to assess LRM capabilities, providing a comprehensive analysis. [TDuK]

**Methodology & Results:**
*   Proposes **two distinct paradigms** (emergent prompting & intrinsic fine-tuning) that effectively enhance metacognition. [DtKq, TDuK]
*   Achieves **"excellent results"** in improving metacognitive abilities, which convinced the reviewer of the method's efficacy. [G22W]

**Writing:**
- Recognize the quality of writing, i.e., **"writing and figures are generally clear"**, **"Quality of the figures is also excellent"**. [fMP3, G22W]

To address the reviewers' concerns and questions, we have provided additional explanations and supporting results in the updated version. The main additions are summarized as follows:

**1. Further Clarification and Validation of Metacognition Ability in Current LRMs:**

* **Self-awareness validation:** We add control analyses on `token length`, `keywords`, `topics` and `privileged self-access` to verify genuine self-awareness, and clarified our *difficulty labels* derived from DeepMath. [fMP3, DtKq]

* **Confidence calculation:** We compare with different confidence calculation metrics (e.g., *median, lowest 10% group*) to rule out outlier effects in confidence monitoring.[G22W]

* **Rigorous comparison:** We clarify the difference between our **task decomposition** setting and traditional CoT scenarios, and conduct detailed ablation studies (*planning v.s. CoT*) to further demonstrate task decomposition act as a multiplier for CoT reasoning. [TDuK]

**2. Further Clarification of Our Designed Metacognitive Reasoning Approaches:**

* **Metacognitive loop necessity:** We add more detailed ablation studies (*w/o planning, confidence monitor, strategy control*) to show the effectiveness of our externalized metacognitive loop framework. [fMP3, TDuK]

* **Comparison with STaR:** We conduct a comparative experiment with **STaR**, demonstrating that our method is *orthogonal* to standard self-correction paradigms. [fMP3]

* **OOD generalization:** We conduct additional evaluations on `GPQA`, `ARC-c`, `MMLU-Pro`, and `LiveCodeBench` to verify the generalization of internalized metacognition framework. [TDuK, DtKq]

**3. Further Explanation of Internal Signal in §3:**

* **Statistical rigor:** We provide additional statistical analysis ($R^{2}$ and $p$-*values*) and *layer-wise probing* analysis to validate the statistical significance and separability of internal metacognitive signals. [G22W]

Thank you for your time again!

---

### Meta-Review · Area_Chair_sBtT · 2026-01-09

**Summary:**

###### Reviewer TDuK

(1) Reservations about the experimental setups and the lack of methodological details provided.

###### Reviewer fMP3

(1) The methodological contributions of the work ("prompting, self-verification, process-supervised fine-tuning, and internal confidence probes") exist in prior work.

###### Reviewer G22W

(1) Latent representations of truth value seem obvious given the nature of neural representations, and prior work; (2) The results of the experiment visualized in Figure 4 are suspect.

###### Reviewer DtKq

(1) The design of the self-awareness study in Section 4 is suspect; (2) Prior and very relevant work in introspection in LLMs and self-awareness is not mentioned; (3) Prior and very relevant work in planning in LLMs is not cited, which undermines the novelty of the experiment in Section 4.3.

**Reviewer Concerns:**

Reviewer fMP3's review does not focus on metacognitive monitoring, so I have downweighted it.

Reviewer DtKq had concerns about the use of the term meta-cognition in the context of the measurements made in this work, which I do not see addressed conceptually by the authors. The remaining concerns, shared by all three reviewers, were in the details of several of the experiments. The authors provided extensive verbose explanations and many new results.

**Reviewer Scores:**

As Reviewer DtKq notes, metacognition is a term borrowed from the philosophy of mind with a rich theoretical history, yet this paper simplifies it into a suite of evals. I do not think that Reviewer DtKq would be satisfied with the strictly results-oriented answer of the author's response to their question.

Further, and more broadly, I am not confident that I can assess that the additional experiments substantively address the shared reviewer concerns about the validity of the experimental setups because of the volume of new material that the authors have added during the rebuttal period. This material should go another round of peer review. Thus, I cannot confidently recommend acceptance.

---

### Decision · Program_Chairs · 2026-01-26

Reject